# Synoptic control over winter snowfall variability observed in a remote site of Apennine Mountains (Italy), 1884-2015

Vincenzo Capozzi, Carmela De Vivo, Giorgio Budillon

Department of Science and Technology, University of Naples "Parthenope", Centro Direzionale di Napoli – Isola C4, 80143, Naples, Italy

*Correspondence to*: Vincenzo Capozzi (vincenzo.capozzi@collaboratore.uniparthenope.it)

**Abstract.** This work presents a new, very long snowfall time series collected in a remote site of Italian Apennine mountains (Montevergine Observatory, 1280 m above sea level). After a careful check, based on quality control tests and homogenization procedures, the available data (i.e. daily height of new snow) have been aggregated over winter season (December-January-February) to study the long-term variability in the period 1884-2020. The main evidences emerged from this analysis lie in (i) the strong interannual variability of winter snowfall amounts, in (ii) the absence of a relevant trend from late 19th century to mid-1970s, in (iii) the strong reduction of the snowfall amount and frequency of occurrence from mid-1970s to the end of 1990s and in (iv) the increase of average snowfall amount and frequency of occurrence in the last 20 years.

Moreover, this study shed light on the relationship between the snowfall variability observed in Montevergine and the large-scale atmospheric circulation. Six different synoptic types, describing the meteorological scenarios triggering the snow events in the study area, have been identified by means of a cluster analysis, using two essential atmospheric variables, the 500-hPa geopotential height and the sea level pressure (both retrieved from the third version of Twentieth Century Reanalysis dataset, which is available for the 1884-2015 period). Such patterns trace out scenarios characterised by the presence of a blocking high-pressure anomaly over Scandinavia or North Atlantic and by a cold air outbreak, involving both maritime and continental cold air masses. A further analysis demonstrates that the identified synoptic types are strongly related with different teleconnection patterns, i.e. the Arctic Oscillation (AO), the Eastern Atlantic Western Russia (EAWR), the Eastern Mediterranean Pattern (EMP), the North Atlantic Oscillation (NAO) and the Scandinavian pattern (SCAND), that govern the European winter atmospheric variability. The relevant decline in snowfall frequency and amounts between 1970s and 1990s can be mainly ascribed to the strong positive trend of AO and NAO indices, which determined, in turn, a decrease in the incidence of patterns associated to the advection, in central Mediterranean area, of moist and cold arctic maritime air masses. The recent increase in average snowfall amounts can be explained by the reverse trend of AO index and by the prevalence of neutral or negative EAWR pattern.

## 1 Introduction

In recent years, the studies focused on past and present snowfall variability have gained more and more attention within the scientific community. It is widely recognized, in fact, that snow exerts a relevant impact within the hydrological cycle,

providing a fundamental reservoir of fresh water, as well as in the whole climate system, controlling the land surface albedo in the global energy budget (e.g. Armstrong and Brun, 2008; Wilson et al., 2010). As well explained by Irannezhad et al. (2017), the occurrence of snowfall events is the results of a delicate balance between thermal and hygrometric conditions and, therefore, it is extremely susceptible to processes and feedback mechanism triggered by global climate change (Berghuijs et al., 2014).

A relevant number of studies reported interesting evidence about last 10-years periods snowfall variability. As an example, less frequent but more severe snowstorms have been observed in the USA (Changnon, 2007). In the Pyrenees region (Spain), Lopez-Moreno et al. (2011) have found an altitude-dependent behaviour of snowfall intensity. After the mid-1970s, a relevant decrease in the number of snowfall days has been detected by Pons et al. (2010) in the northern part of the Iberian Peninsula; according to the authors findings, such evidence is the result of an increase in temperature at low elevations and of a decrease in the precipitation recorded at high altitudes. A further focus on the Iberian Peninsula, and in particular on the Castilla y Leon region, was provided by Merino et al. (2014), which highlighted a negative trend in snowfall days for the period 1960-2011, as well as interesting linkages between regional snowfall variability and large-scale atmospheric patterns. Focusing on the Alpine region, Micheletti (2008) carried out an analysis for the Friuli Venezia Giulia (northeast of Italy) for the 1972-2007 period, by investigating the seasonal depth of snowfall records from eight stations. A positive anomaly was observed until the end of 1980s, followed by a lowering of snowfall amount between 1990 and 2000 and by a subsequent recovery (but below the level of 1980s). Valt and Ciafarra (2010) investigated the accumulated snowfall variability observed in Italian Alps (eastern and western sectors) in the period from 1960 to 2009. As general results, they found a negative trend, which is strongest in spring season and below 1500 m above sea level (hereafter, asl). Terzago et al. (2010) have provided a focus on the Piedmont Region (northwest of Italy), examining the monthly snow depth and depth of snowfall for the period 1971-2009; the authors discovered an increasing trend for November-December period and a decreasing tendency in January-April. Kreyling and Henry (2011) analysed 177 stations in Germany for the period 1950-2000, focusing on snow days variability: they found a negative trend for the majority of the stations, especially in the last 15 years of the considered period. Marcolini et al. (2017) investigated the snow depth time series collected in the Adige basin (North East Italy) for the period 1980-2019. According to the results of this study, a relevant reduction of both snow cover duration and mean seasonal snow depth occurred in the Adige catchment after 1988, both at low and high elevation sites. On the contrary, in the period from 2000 to 2009, an increase (although not identified as statistically significant) of mean seasonal snow depth has been recorded. Lejeune et al. (2019) found a relevant decrease in the snow depth observed in Col de Porte (France) during the period 1960-2017. Moreover, it is worth mentioning the study of Schöner et al. (2019), which examined a very high number (139) stations located in Austria and Switzerland for the period 1961-2012. The authors of this work found negative tendencies in snow depth (up to -12 cm / 10 years period for sites at elevations of about 2000 m asl) and a positive relationship between the strength of snow depth trends and the altitudes. Very recently, Matiu et al. (2021) have presented a comprehensive assessment of snow depth trends in the European Alps. They observed negative trends in snow cover in the period from 1971 to 2019, especially in the spring season.

Moreover, this study highlights that mean snow depth in the Alpine region has a very strong interannual variability and that it exhibited a very strong decline in the 1990s, followed by a partial recovery from 2000s.

The study period of the climatological researches carried out in such works is generally limited to the last 55-65 years. However, few studies (mainly focused on the Alpine area) extended their analysis further back, due to the absence, in many areas, of reliable and continuous old snowfall climatological records. In this respect, it is deserving of mention the study of Scherrer et al. (2013), which investigated the snowfall variability observed in Switzerland during the 1864-2009 period, using nine different stations. According to the findings of this work, the analysed depth of snowfall time series exhibit a strong decadal variability. The highest value in depth of snowfall and days with snowfall occurred in 1900-1920 and 1960-1980 period, the lowest between 1980s and 1900s, whereas an increase in depth of snowfall has been observed in 2000s. Another important reference for mountainous areas is Laternser and Schneebeli (2003), who discovered, for the Swiss Alps, an increase in snow cover and duration from 1930 to early 1980 and, subsequently, a statistically significant decrease towards 1999. Some years later, Beniston (2012b) observed a decline in winter snow depth in the 1930-2010 period by analysing 10 time series in Switzerland. The study of Marty and Blanchet (2012), which has investigated a very large number of snowfall records collected in Switzerland during 1931-2010 period, reached similar conclusions. The authors discovered that the 44% of the considered stations show a significant decrease in the annual snow depth maxima values. Another interesting evidence can be found in the study of Schöner et al. (2009), which examined the snow depth time series collected in Sonnblick (Austria) in the 1928-2005 period: the authors reported a strong interannual variability in snow depth (with largest values in 1940-1950 period) and a decreasing trend in summer snow. Terzago et al. (2013) found a decrease in snow depth (2-14 cm per decade) in the western Italy for the period 1926-2010. Moreover, they also reported a very strong interannual variability, with maxima in 1940, 1950 and 1960, minima in 1990 and then a recovery in the 2000s. Irannezhad et al. (2017) evaluated the annual snowfall variability in Finland in the 1909-2008 period, analysing its relationship with some large-scale atmospheric patterns. From this study, it emerges a significant decline in annual snowfall in the northern part of Finland, as well as a strong relationship between the observed snowfall variability and the large-scale teleconnection patterns synthesised by Arctic Oscillation and East Atlantic indices.

This study aims to provide a new contribution to this research path, by analysing the snowfall data collected in the last 137 years (1884-2020) in a remote site of Southern Apennine Mountains, Montevergine observatory (40.936502° N, 14.729150° E), located in Campania region at 1280 m asl.

The main purposes of our work can be synthetized as follows:

- To extend, from both quantitative and qualitative perspectives, the current knowledge about the past snowfall variability observed in Mediterranean mountainous sectors;
- To shed light on links between large-scale atmospheric circulation and local climate variability, by identifying and analysing the synoptic patterns favourable to winter snowfall events in Montevergine as well as their relationship with the main teleconnection indices that govern the atmospheric circulation in the Mediterranean area.

As well highlighted in some previous studies (Diodato, 1992; Capozzi and Budillon, 2017; Capozzi et al., 2020), the
Montevergine climatic series offers a relevant opportunity to meet these goals. It has, in fact, some distinguishing features,
mainly related to its longevity (it is the oldest collected in the Apennine region above 1000 asl) and to the environment in
which it has been collected, i.e. a remote mountainous area of Central Mediterranean region.
This paper has been structured as follows. Section 2 presented the Montevergine snowfall time series, describing the procedures
adopted for quality control and homogenization; moreover, it provides details about data and methodologies used to establish
the synoptic scale patterns, as well as a brief description of the teleconnection indices involved in this study. The results are
sketched in section 3, whereas section 4 is dedicated to the relationships between the identified synoptic types and
teleconnections. Finally, the discussion and the concluding remarks are provided in Section 5 and 6, respectively.
**2 Materials and methods**
**2.1 Study area and snowfall data**
The snowfall climatological time series involved in this study has been collected in Montevergine Observatory (hereafter,
MVOBS). The latter is located in Campania Region (Southern Italy) on the eastern side of the Partenio Mountains, which are
part of the Southern Apennines (Fig. 1, left panel). As clearly highlighted in some previous papers (e.g. Capozzi et al., 2017;
Capozzi et al., 2020), MVOBS has a very complex meteorological regime, modulated by a large spectrum of synoptic patterns.
Its geographical position, in fact, barycentric with respect to the Mediterranean Sea and Balkan regions, results in the frequent
incidence of moist air masses coming from near Atlantic and Mediterranean basins as well as of cold outbreaks coming from
Northern and Eastern Europe. Both weather types are associated with precipitation events, which are sometimes emphasised
by local orographic features. Considering the last climatological reference period (1981-2010), MVOBS receives an average
annual precipitation of 1606.8 mm ($\pm$ 338.0 mm), with maxima average amounts in winter (527.8 $\pm$ 191 mm) and fall (518.5
$\pm$ 186 mm) seasons, and a minimum in summer (163.0 $\pm$ 88 mm). It is important to point out that if we consider the entire
observational period, which spans from 1884 up to date, the annual average precipitation rises to around 2100 mm. As it will
be discussed in section 3, a strong and relevant reduction of precipitation amounts has been recorded between the mid-1970s
and late 1990s.
The meteorological observation in MVOBS started on January 1, 1884, under the management of the "Italian Central Office
of Meteorology and Geodynamics" (hereafter, the Italian Central Office, ICO), which was established in Rome in 1879
(Maugeri et al., 1998). From the first years, the observatory was equipped with many instruments, most of them placed on a
meteorological tower, built up on the eastern side of the Montevergine Abbey in 1895 (Capozzi et al., 2020). According to the
available metadata, the nivometric measurements started on 01 April 1884; in compliance with the recommendations of the
ICO, the snowfall measures have been performed manually using a traditional nivometer. The latter consists of a snowboard
and of a graduated yardstick with an area of 0.01 m$^2$ and a section of 0.0001 m$^2$, respectively. The snow observations involve
two essential variables, the height of new snow (i.e. the amount of fresh snow with respect to the previous observations,

hereafter HNS) and the snow depth (i.e. the vertical distance from the snow surface to the soil level, hereafter SD). The measurements have been collected in the "*Giardinetto dell'Ave Maria*", a cloister located next to the northern side of the meteorological tower.

MVOBS snowfall data include both daily and sub-daily observations. More specifically, in this work we used the daily HNS (hereafter, $HNS_d$) measurements, which cover the entire analysed period (from 1884 up to 2020) with some few gaps mainly located in the 1960s (1964-1968), when the observatory was closed due to the personnel lack. It is important to highlight that we are not able to reconstruct the missing data due to the unavailability of snowfall time series collected in sites close to Montevergine. Additional details about data availability and source and about measurements practises can be found in the Appendix A.

From $HNS_d$ data, we have computed the monthly HNS values by a simple sum of all $HNS_d$ data observed in a determined month. It is important highlighting that we have calculated the monthly values if 95% of daily observations are available. The total winter HNS value has been determined, for a certain year, by adding the monthly HNS values recorded in December (previous year), January and February.

The entire dataset, originally available on hand-written meteorological registers, has been transcribed into a digital support by the Department of Science and Technology of the University of Naples "Parthenope", using a simple "key entry" approach (Brönnimann et al., 2006). A part of the dataset, including the sub-daily meteorological data collected between 1884 and 1963, is publicly available from the National Oceanic and Atmospheric Administration (NOAA) - National Centers for Environmental Information (NCEI) repository (Capozzi et al., 2019).

It is worth mentioning that manual snowfall measurements have several limitations, which are well summarised and discussed in WMO (2008). Such uncertainties rely on the observer's errors in reading and recording the measure and in poor siting. The use, as in the context of Montevergine, of a single stake could result in large errors, depending on the site mean variability: according to WMO (2008), this error increases with decreasing snow depth. The limitations related to the siting area may be exacerbated by environmental conditions, such as strong winds and/or turbulence, which may generate snow drifts and spatial inhomogeneity in snow depth. According to verbally transmitted information (provided to the authors of this paper by the Benedictine Community of Montevergine), to mitigate the occurrence of poor siting conditions (mainly caused by strong winds), the snow depth value taken from reference observation (collected with a fixed stake) was replaced by an average snow depth value coming from spatially distributed non-fixed observations. Unfortunately, the use of this alternative measurements method has not been systematically mentioned in the meteorological registers or in metadata.

## 2.2 Quality control and homogenization

It is well known that the consistency and the reliability of an historical climatological time series can be undermined by several errors and artefacts, dealing with human mistakes in data collection, inaccuracies in digitization procedures and instrument

failures. For this reason, as highlighted by many papers (e.g. Ashcroft et al., 2018), it is essential applying a Quality Control (QC) procedure to identify both systematic and non-systematic errors.

However, in meteorological and climatological fields, there is not an unambiguous and universal QC strategy. Depending on the number of data, in fact, different methods can be employed. As an example, in a scenario involving a large dataset including high-resolution data, it is possible to opt for methods that allow not only identifying anomalous values, but also to devise criteria for correcting them. On the other hand, for a single isolated time series, such as the one considered in this work, it is convenient to select simpler methods that are able to accept or reject a determined observation depending on statistical objective criteria.

In this study, following the guidelines provided by WMO (2011) and by the Istituto Superiore per la Protezione e la Ricerca Ambientale (ISPRA), we developed a QC strategy based on two different statistical test, the gap check and the climatological control (ISPRA, 2016). Both tests are designed to detect the outliers, i.e. the anomalous data that exceed specific limits defined according to an objective probability distribution model or to specific climatological values.

More specifically, the gap check detects unrealistic breaks in the frequency distribution of a temperature or precipitation data sample. We have applied this method as follows. Firstly, for each month we have sorted in ascending order the $HNS_d$ data recorded over the entire available period (1884-2020). Subsequently, we have calculated the difference between two-consecutive values: if a certain $HNS_d$ value is at least 35 cm larger than the previous record, then that value and subsequent ones are flagged as outliers. In the selection of the just mentioned threshold (35 cm), we followed the guidelines and the suggestions provided by NCEI of NOAA (Durre et al., 2010). As testified by Fig. 2, which shows the frequency distribution of $HNS_d$ for winter months (December, January and February), no outliers were detected by gap check.

The second procedure, the climatological control, checks for $HNS_d$ values that go beyond the respective 29-days climatological 95th percentiles by at least a certain factor (which is equal to 5 according to the recommendations of NOAA's NCEI). We have applied this method to each calendar day, considering the entire available time series (1884-2020). The results confirm the very encouraging findings of gap check: no outliers, in fact, have been detected.

In addition to quality control, it is common practice to apply statistical methods to climatological observations in order to verify the time series homogeneity and identify the possible presence of artificial shifts or trends that can affect the results of data analysis (Wijngaard et al., 2003; Hänsel et al., 2016). More specifically, the inhomogeneity in climatic data can be attributed to several factors, including instrumentation error, changes in the environment adjacent to the instrument and human mishandling.

We applied two methods to detect and locate abrupt changes in the snowfall time series involved in this study, namely the Pettitt test (Pettitt, 1979) and the CUSUM test (Smadi et Zghoul, 2006; Hänsel et al., 2016; Patakamuri et al., 2020; Gadedjisso-Tossou et al., 2021).

The two homogeneity tests were performed using the "Anabasi 2018 version 1.51 beta" (ANAlisi statistica di BAse delle Serie storiche di dati Idrologici), a specific tool developed by ISPRA in the Visual Basic for Application language (Braca et al.,

2013). We applied both tests to the monthly HNS climatological series. The results obtained from the application of Pettitt and
CUSUM tests through the Anabasi tool did not reveal any change points in the historical series.

**2.3 Reanalysis data**
The reanalysis data represent a very commonly used product in studies concerning the examination of atmospheric circulation
patterns and their correlation with local climate variability (e.g. King and Turner, 1997; Cohen et al., 2013).
In this work, we used the Twentieth Century Reanalysis dataset (20CR), which has been generated by the NOAA and by the
University of Colorado Boulder's Cooperative Institute for Research in Environmental Sciences (CIRES), in partnership with
the International Atmospheric Circulation Reconstructions over the Earth Initiative (ACRE; Atlas et al., 2011). More
specifically, we selected the latest version of this dataset (the version 3), which has been launched by NOAA, CRIES and the
U.S. Department of Energy (DOE). The NOAA-CIRES-DOE 20CR Version 3 (20CRV3) provides a four-dimensional global
atmospheric dataset spanning from 1836 to 2015 at 1.0° latitude/longitude resolution. Main advances of this product with
respect to the previous versions lie in the improved data assimilation system, in the assimilation of a large set of observations
and in the usage of newer and higher resolution models (Slivinski et al., 2019).
To characterise the synoptic conditions that triggered the snow events in the site of interest, we focused on two parameters:
the 500-hPa geopotential height (hereafter, Z500) and the sea-level atmospheric pressure (hereafter, SLP). The first one can
be considered a "primary" meteorological field, giving essential information about large-scale mid-tropospheric flow as well
as about the location and intensity of ridges, troughs and sub-synoptical patterns (i.e. upper level lows). The SLP is a
"secondary" atmospheric field, providing information about location and intensity of main surface synoptic features, such as
low and high-pressure systems, which can be regarded as the result of dynamical and thermodynamic processes that involve
the entire tropospheric column.
For the purposes of this work, we extracted from 20CRV3 the daily (00:00 UTC) Z500 and SLP data for the winter seasons
(01 December-28 February) of the 1884-2015 time period, considering an area that includes the entire European territory (25-
90°N, -45°W-65°E). This region appears to be reasonably representative of the synoptic conditions that control the
meteorological regime of the site of interest. The dimensions of the domain were selected to be consistent with the synoptic
scale analysis of the present work and to avoid circulation features in regions remote from the study area (Merino et al., 2014).
**2.4 Cluster analysis**
Among the different methodologies and approaches usually involved in synoptic types classification, we have selected the
cluster analysis (CA). The latter is an unsupervised learning technique that aims at finding natural grouping and pattern in a
determined data set. The literature offers several example of application of CA (e.g. Kidson, 1994a; Kidson, 1994b; Kidson,
2000; Cohen et al., 2013), involving some essential synoptic variables, such as the mean sea level pressure field and the 1000,
700 and 500-hPa geopotential heights.
In this study, we have applied the CA as follows. Firstly, using the available 20CRV3 reanalysis data, we have computed the
Z500 and SLP anomalies, considering the 1981-2010 as climatological mean reference. Subsequently, we have selected the
wintertime days in which snowfall precipitation has been observed in MVOBS. In this respect, it is important highlighting that
in our study a "snow day" is a day on which accumulated snowfall (i.e. $HNS_d$) is at least 3.0 cm. This threshold allows filtering
out most of some "ambiguous" events, characterised by the simultaneous presence of different hydrometeors types (i.e. rain,
snow hail or graupel). Using this criterion, the time-dimension of the dataset, originally including 50039 time points (i.e. all
winter days between 1884 and 2015), reduces to 1986 days. Prior to clustering, the selected data were subjected to a Principal
Component Analysis (PCA). This approach is commonly adopted (e.g. Yeung and Ruzzo, 2001) in order to reduce the
dimensionality of the data set and, consequently, the computational cost of CA. For both parameters (Z500 and SLP), we have
considered the first eight PCA components; the latter, according to the scree plots (Fig. 3), explain over the 80% of the total
variance (more specifically, the 83.8% for Z500 and the 89.0% for SLP). After this pre-processing, we have applied the well-
known k-means method (MacQueen, 1967) using as input the 16 components resulting from PCA. Following standard
procedures to assess cluster reproducibility and stability, we have performed several sensitivity experiments, by running the
k-mean procedures using different randomly selected seed clusters. The final selection of the optimal number of clusters has
been performed by manual inspection. According to this strategy, six different clusters were extracted: for each of them, we
determined the frequency of occurrence for every winter season, as well as other basic statistical indicators, such as the
associated average and standard deviation of $HNS_d$ values.
The synoptic type classification is a powerful and descriptive tool for summarising the typical modes of atmospheric circulation
that affects local climatic features. However, it should bear in mind that the identification of discrete categories presents several
limits, mainly related to (i) the challenges in identify the boundaries between synoptic classes (i.e. patterns may switch
abruptly, but in some occasion there is gradual transition from a pattern to another one), (ii) the impact of minor features, such
as weak low or transitory ridge, that may not negligible in some circumstances and (iii) to the seasonal variation in type
characteristics (i.e. synoptic features are generally most relevant in winter).
**2.5 Teleconnections**
To investigate the relationships between large-scale climate variability and the synoptic types identified in this study, we have
considered five different teleconnections: the Arctic Oscillation (AO), the East Atlantic Western Russia (EAWR), the Eastern
Mediterranean Pattern (EMP), the North Atlantic Oscillation (NAO) and the Scandinavian Pattern (SCAND). These large-
scale modes are known to have relevant influences on the atmospheric circulation of the Mediterranean area, especially in
winter. For the convenience of the reader, we provide below a brief description of each teleconnection:

- AO: it is one of the leading modes of the northern hemisphere climate variability, describing the circulation pattern over the mid-to-high latitudes. The positive phase of this mode is associated with a ring of strong winds circulation around the North Pole and is synonymous with a vigorous polar vortex, confining the cold air across northern latitudes. When a negative phase occurs, the belt of winds become weaker and the polar front is subjected to relevant oscillations, resulting in cold air masses outbreaks across mid-latitudes.

- EAWR: it is a prominent teleconnection pattern that affects Eurasia throughout the year (Barnston and Livezey, 1987). It consists of four main Z500 anomaly centres, placed in sequence from west to east in the latitude range between 40 and 70°N. The EAWR positive phase is characterised by positive Z500 anomalies over Europe and northern China, and negative height anomalies over the central North Atlantic and north of the Caspian Sea. In the negative phase, the poles reverse and, therefore, Europe experiences negative Z500 anomalies.

- EMP: this pattern has been identified by Hatzaki et al. (2007) and it is defined as the difference in Z500 between the north-eastern Atlantic and the eastern Mediterranean. The positive phase is associated with a positive Z500 anomaly in the north Atlantic and to a negative Z500 anomaly across the central and eastern Mediterranean area. This Z500 anomalies placement means, in winter season, favourable conditions to cold air outbreaks over central and southern Europe. Oppositely, the negative EMP phase means positive Z500 anomaly over the eastern Mediterranean and northern Africa, where an anticyclonic circulation predominates.

- NAO: it can be considered the Atlantic branch of the AO and it describes the sea-level pressure pattern between the Northern Atlantic Ocean (near Greenland and Iceland), where generally the air pressure is lower than surrounding regions, and the Central Atlantic Ocean (near Portugal and Azores), which generally experiences air pressure higher than surrounding areas. The NAO positive phase means that both subpolar low and the subtropical high are stronger than average and that, consequently, the Atlantic jet stream is confined to high latitudes. The negative phase results in a weakening of both subpolar low and subtropical high and, therefore, in a southward shift of the storm track.

- SCAND: this pattern consists of a primary relevant circulation centre over Scandinavia and weaker poles of opposite sign over Western Europe and eastern Russia/western Mongolia. The positive phase, characterised by positive Z500 anomalies over the Scandinavia region and western Russia, reflects synoptic scenarios that promote the formation of blocking anticyclones. During the opposite phase, negative Z500 anomalies prevail over the Scandinavia area and weak positive anomalies over Western Europe.

The AO, EAWR, EMP, NAO and SCAND indexes have been retrieved from the Climate Prediction Center (CPC) of the NOAA's National Weather Service, where monthly data from 1950 are available (https://www.cpc.ncep.noaa.gov/data/teledoc/telecontents.shtml). For each index, we have extracted the winter season time series by simply averaging, for a determined year, the monthly values of December (previous year), January and February. Regarding the EMP, its variability was ad-hoc reconstructed using the reanalysis data involved in this study (20CRV3 product), following the methodology suggested in Hatzaki et al. (2009). The EMP has been firstly reconstructed, for the 1950-2015 period, on a monthly basis; subsequently, the data have been standardised with respect to 1981–2010 climatology and averaged

on winter season, in order to have a time series that is fully consistent with the ones available for the other teleconnection
indices.

## 3 Results

### 3.1 Snowfall variability in 1884-2020 period

In this section, we present the evidence provided by the snowfall time series collected in MVOBS between 1884 and 2020.
Figure 4 presents the results in terms of total HNS (left panel) and number of snowy days (right panel) observed in winter
season year-by-year (blue bars). To filter out the high-frequency variability, we have applied a moving average smooth,
highlighted as a red curve, setting the time window equals to 20 years. The missing data are highlighted as yellow bars.
Moreover, to better emphasise the interannual variability, we have also computed the moving window standard deviation
(setting a 20-year window), which is shown as magenta line in the bottom panels of Fig. 4.
A simple visual inspection of total HNS signal allows identifying some relevant features: (i) the strong interannual variability,
which is typical of precipitation records collected in mid-latitudes, and (ii) the strong reduction in snowfall amounts between
mid-1970s and late 1990s. In order to examine the behaviour of winter HNS in MVOBS, we have subdivided the time series
into seven different sub-periods, having a 20-year length (except for the last one): 1884/85-1903/04, 1904/05-1923/24,
1924/25-1943/44, 1944/45-1963/64, 1964/65-1983/84, 1984/85-2003/04 and 2004/05-2019/20. Table 1 shows, for each time
segment, the average HNS value and the associated standard deviation. The first two sub-periods have the highest total HNS
average among all considered time intervals. They exhibit similar values not only in terms of mean (213.2 and 221.5 cm,
respectively) but also of standard deviation (± 104.3 cm and ± 103.6 cm, respectively). The moving average standard deviation
is quite stable and ranges approximatively between 100 and 120 cm. Subsequently (1924/25-1943/44), the average HNS value
drops to 199.7 cm, due to a considerable but transitory reduction of snowfall amounts occurred in the first part of 1920s; no
relevant variations in terms of moving standard deviation can be observed. Among mid-1940s and mid-1960s, there are no
important changes in average HNS, which slightly rises to 201.3 cm, whereas the standard deviation (± 140.1 cm) is higher to
that observed in the previous segments. This rise in standard deviation can be explained by the occurrence of some seasons
(1953/54 and 1955/56) characterized by excessive amounts (579 and 514 cm, respectively) as well as of meagre winters
(1954/55 and 1963/64), in which the accumulated HNS was 16 and 30 cm, respectively. The very pronounced interannual
variability detected in this period is well synthetized also by the moving average standard deviation, which is between 130 and
150 cm. It should be noted that the statistics of this period might be adversely affected by some missing data. In the following
two sub-periods (1964/65-1983/84 and 1984/85-2003/04), a gradual and impressive decrease in HNS amount has been
detected. The strong anomalies observed in these periods are well emphasised not only by the average HNS value (which drops
to 114.1 cm between mid-1980s and mid-2000s), but also by the moving average standard deviation, which declined to around
60-80 cm between mid-1970s and the end of 1990s. According to Table 1, in the last time interval there was an increase in the
average HNS amount (which rises to 167.4 cm), although it is quite lower than that generally observed before the 1970s. The
standard deviation returned on values very similar to those detected in the first four time segments, instead.
Additional and relevant information about the variability of snowfall regime observed in MVOBS are provided by the right
panel of Fig. 4, which shows the behaviour in time of the number of snow days (hereafter, NSD) occurred in winter between
1884/85 and 2019/20 seasons. In the first four sub-periods, the average NSD value fluctuates between 16.1 and 20.3 days and
the standard deviation ranges from ± 7.4 to ± 9.1 days. Subsequently, in alignment with that observed for HNS, in the period
from mid-1960s to mid-2000s there was a gradual reduction in snowfall events frequency, which lowered to 11.9 days (± 5.7)
in the 1984/85-2003/04 sub-period. The absolute minimum NSD value was observed in the 1989/1990 season, when only 2
snowfall days occurred. The similarities with the evidence provided by the analysis of total winter HNS also rely on the
reduction of NSD in the 1920s and in its slight recovery after the 2000s.

**3.2 Synoptic types**

In this section, we discuss the six synoptic types (ST) emerging from CA. Each pattern represents a typical synoptic scenario
that is able to trigger snowfall events in MVOBS. For the convenience of the reader, we labelled each cluster as follows: the
first type ST1, the second one ST2, the third one ST3, the fourth one ST4, the fifth one ST5 and the sixth one ST6.
Figure 5 shows the first three synoptic types, ST1, ST2 and ST3. More specifically, the left panels present, for each ST, the
Z500 anomaly (in m), whereas the right panels the corresponding SLP anomalies (in hPa).
The ST1 shows a synoptic condition that promotes the incoming of cold arctic maritime air masses in the central Mediterranean
area. This pattern is typically the fruit of a relevant oscillation of the polar front, triggered by a positive Z500 anomaly (Fig.
5a) over the northern Atlantic, between Greenland and Iceland. In most of European territory, a trough elongated from northern
to southern Europe conditions the atmospheric circulation. The latitudinal extension of negative Z500 anomaly suggests that
this pattern represents, on average, the mature stage of the Rossby wave associated with the polar front oscillation, which
results in strong thermal and pressure gradients. Because of this upper level circulation, the low-level conditions are modulated
by two opposite poles (Fig. 5b), a positive SLP anomaly (up to 15 hPa) over the northern Atlantic and a negative SLP anomaly
over eastern Europe, Balkans and Central and Southern Italy. The scenario just described leads to the development of a strong
baroclinicity in the area of interest, where very favourable conditions for snow events occur.
The ST2 (Fig. 5c,d) presents a synoptic situation modulated by a relevant negative Z500 anomaly over Central Europe (located,
on average, between Eastern of France, Alpine region and Southwestern of Germany). This upper level anomaly can be
considered the final stage of a Rossby wave that brings cold arctic maritime air masses from northern to central and southern
Europe. This negative Z500 anomaly is counteracted by a positive anomaly over Greenland, north-western Atlantic and Arctic.
This pattern results in a cyclonic circulation over the Central Mediterranean area, well represented by a negative SLP anomaly
on the Central Italian peninsula, which triggers a cold and moist southwestern flow over the area of interest.

The ST3 resembles a synoptic scenario that promotes the incoming in central Mediterranean basins of very cold continental air masses, which can be classified in continental arctic, when the air masses originate in northern of Russia and Novaya Zemlya, and in continental polar, when they come from Siberian plains. Focusing on Z500 anomaly (Fig. 5e), it is easy to recognize a synoptic configuration that strongly departs from the classical schemes followed by atmospheric circulation in the European area. The main features of this anomalous scenario lie in the presence of a high-pressure area extended from mid-Atlantic to Scandinavian Peninsula with a north-eastern to south-western axis. Moreover, there are two negative Z500 anomalies, one located over Greenland and Iceland and the other one across southern Italy. The latter is associated with a low-pressure area, located, on average, on the Ionian Sea (Fig. 5f). According to this pattern, the central and eastern European territories experience stable but very cold weather, whereas southern Italy is affected by cyclonic conditions that drive a north-eastern flow over the area of interest. Although the continental air masses are in general drier than maritime ones and so less prone to force clouds and precipitation formation, this synoptic set-up still supports snowfall events in the study area. In this respect, two important factors must be taken into account: the passage of cold continental air masses over the Adriatic Sea, which induces vertical transport of moisture and heat that enhance the instability, and the orographic forcing that comes into play when the air masses interact with Partenio Mountains.

The ST4 depicts a large-scale flow conditioned by two relevant Z500 anomalies: one, of negative value, located across western and central Mediterranean basins, and the other one, of opposite sign, located over high latitudes (greater than ≈55°N) from northern Sea to Western Russia. This scenario, similarly to ST2, can be considered the result of a pronounced jet stream oscillation, bringing cold arctic or polar continental air masses towards southern Europe, and represent the final stage, i.e. cut-off low, of the Rossby wave evolution. The synoptic configuration sketched in Fig. 6a and Fig. 6b is synonymous with strong baroclinic conditions over central and southern regions of southern Italy and supports relevant snowfall events in the study area.

The ST5 shows a mid-tropospheric scenario modulated by a wide trough, extended on European central meridians from Scandinavian Peninsula to central Mediterranean basins, and by a strong positive Z500 anomaly over mid-Atlantic (Fig. 6c). This pattern leads to cold arctic advection towards the Italian peninsula, mainly of maritime origin. From the analysis of SLP pattern (Fig. 6d), three different pressure anomalies can be easily detected: a positive one over the Atlantic Ocean (which is compatible with a blocking high-pressure area) and two negatives, one over northern Europe and the other one across southern Italy and Balkan Regions. The meteorological set-up synthetized by ST5 is favourable to cyclonic conditions over the study area, which receives precipitation amounts mainly of convective nature (caused, for example, by the passage of a cold front, a comma or a cold occlusion).

The first five ST discussed so far have some commonalities, mainly lying in the presence of a ridge over northern and central Atlantic (i.e. blocking high conditions), which results in very cold outbreak over most part of Europe. The ST6 depicts a very different pattern, which supports the incoming, over western and central Europe, of polar maritime air masses. By the inspection of Fig. 6e, four different Z500 anomalies can be detected: two of negative sign, located west of British Islands and over central Italy, and two of positive sign, situated over northern Europe and western Russia and south-western of Iberian

Peninsula, respectively. The SLP anomalies (Fig. 6f) resemble the mid-tropospheric configuration and support cyclonic conditions over the study area, associated to convective precipitation events mainly triggered by sharp contrasts between cold air coming from north-western and milder Mediterranean air masses.

According to Table 2, the average frequency of occurrence (computed over the entire analysed period, 1884-2015) is quite homogeneous among various ST, i.e. there is not a dominant ST. In particular, ST4, ST6 and ST1 have, on average, the higher frequency of occurrence, 19.8, 19.3 and 17.7%, respectively. The remaining three patterns occur slightly less, but almost equally (12.1-16.3%). Moreover, we have averaged the amount of $HNS_d$ values recorded in the events triggered by each ST. The results, presented in Table 2, show that snowfall events are heavier when they are forced by a synoptic scenarios ascribable to ST4 (12.6 ± 10.8 cm), ST2 (12.0 ± 10.3 cm) and ST1 (11.8 ± 10.4 cm). The other three clusters (ST3, ST5 and ST6) are associated with average $HNS_d$ values slightly lower (9.9, 9.7 and 10.2 cm, respectively). It should be pointed out that the average $HNS_d$ found for ST1, ST2 and ST4 significantly differ from the $HNS_d$ values found for ST3, ST5 and ST6 (i.e. according to the t-test, $p<0.05$ for all ST pairs, for example ST1 vs. ST5 or ST2 vs. ST6).

## 3.3 Variability in time of synoptic types

The information about the frequency of occurrence of the six ST have been extracted as follows. For each winter season of 1884-2015 period, we have identified the NSD associated with each cluster. The results are sketched in Fig. 7, where the temporal evolution of a determined ST frequency of occurrence is represented by the blue line and is expressed as number (#) of days per winter season. In order to filter out the high frequency variability, we have applied a moving average smoothing (red line), using a time window of 20 years. From a simple visual inspection of the six panels of Fig. 7, it clearly emerges that none of the STs exhibit a defined linear trend, except for ST1 and ST2, whose frequency of occurrence is affected by a negative tendency although not statistically significant. It is important to point out that the statistical significance of the trends has been evaluated through the popular Mann-Kendall test (Mann 1945, Kendall 1962).

Starting from ST1 (Fig. 7a), its incidence on snowfall events in MVOBS was quite constant (with an average of about 4.0 NSD per winter season) until the mid-1910s, then dropped to about 2.0 NSD between 1920s and 1930s. Subsequently, the number of snow events associated with this pattern rose to about 3.0 NSD per winter season and remained quite stable until the 1960s. A gradual decrease of the ST1 occurrence can be easily detected between the 1960s and 1980s. In the 1983/84-2003/04 period and in the last time segment (that ends up to 2014/15 season for this analysis), no relevant variations of average incidence of this ST have been detected. The behaviour in time of ST2 frequency of occurrence (Fig. 7b) can be outlined as follows: a strong interannual variability in the first time segment, a decline in second one (mainly in the 1910s), a gradual recovery from 1930s to 1960s, then a new drop culminating in the 1990s, when the incidence of ST2 on MVOBS snowfall regime was very low (≈ 1.0 NSD per year, on average) and, finally, a slight increase in the last sub-period.

The ST3 (Fig. 7c) incidence on snowfall events in MVOBS decreased in the fourth sub-period (i.e. 1964/65-1983/84) and then
it was quite stable up to the end of the investigated time interval. A temporary lowering can be observed only between the end
of 1970s and the first 1980s. The ST4 frequency of occurrence (Fig. 7d) exhibit a negative trend in the first two sub-periods,
a recovery in the period from 1930s to 1950s and then a decrease in the fourth time segment. Subsequently, the time evolution
of ST4 was modulated by a strong interannual variability; a greater incidence of this pattern on MVOBS snowfall events has
been detected in the last fifteen years, when its frequency of occurrence was 2.8 events per year. The ST5 had a high frequency
of occurrence in the first years of 20th century, and then its incidence rapidly dropped and remained almost stationary until the
end of the analysed period (Fig. 7e). The ST6 has the following behaviour: in the first 20 years, it exhibited a rising trend,
maximising its incidence in the 1910s, and then its frequency declined and remained quite stable until the early 1980s.
Subsequently, its impact on MVOBS nivometric regime reduced between 1980s and 2000s, and then it has raised in the last
15 years.
Table 3 presents a linear correlation analysis between the ST frequency of occurrence and the total winter HNS signal shown
in Fig. 4. The correlation coefficient values (hereafter, $r$) marked in bold indicate correlations with 90% significance levels,
according to the p-value test. Starting from the first time interval, a very strong correlation has been found for ST1, ST2 and
ST4. In the 1904/05-1923/24 period, the total HNS times series has a relevant connection with ST4 variability ($r = 0.58$), with
ST2 ($r = 0.45$), as well as with ST3. The subsequent period was strongly modulated from ST1, ST2 and ST4; a moderate
correlation has also been found for ST3. In the time interval from 1940s to 1960s, the highest correlation levels have been
identified for ST2, ST3 and ST4 ($r = 0.63$, $r = 0.46$ and $r = 0.78$, respectively). In the successive two time intervals, when a
strong reduction in HNS amount occurred, the most relevant correlation has been found with ST1, ST5 (only in 1964/65-
1983/84 period), ST4 and ST2 (the latter only in 1984/85-2003/04 period). The behaviour of HNS amount observed in MVOBS
in the last 15 years has a direct linkage with ST4 ($r = 0.75$), ST3 ($r = 0.62$) and ST1 ($r = 0.58$) patterns. Considering the whole
investigated time interval (i.e. 1884/85-2014/15), the highest correlations have been found for ST4 ($r = 0.60$), ST1 ($r = 0.56$)
and ST2 ($r = 0.52$).
The key-findings of this analysis may be summarised as follows. In the period between 1970s and 1990s, when a strong
reduction of snowfall events was observed in MVOBS, there was a relevant decrease of ST2 and ST1 frequency of occurrence.
The ST3 and ST6 also exhibited a lowering in their incidence in this period, although less prominent, whereas the ST4 and
ST5 did not show any particular variations in their tendency, although they have a good correlation with the behaviour in time
of winter HNS amounts. The recovery in snowfall amounts and number of events observed in MVOBS in the most recent sub-
period can be related to the rising in frequency of ST4, ST2 and ST6. However, referring to the last interval, the ST1 and ST2
incidence remains relatively low, in absolute terms, if compared to that observed prior to the 1970s.
Figure 8 provides further evidence about variability in time of the ST analysed in this study. It shows, for each of the seven
time intervals, the frequency of occurrence of ST relative to the total number of observed snow events. It is very interesting
focusing on the integrated contribution of ST1 and ST2 (blue and red bar, respectively). In the first four time segments, it
ranges from 26% (1904/05-1923/24 period) to 39% (1944/45-1963/64 period), then it strongly decreases up to 23% in the
1984/85-2003/04 period and to 22% in the last sub-interval. At the same time, it is worth mentioning the increase, over the
1984/85-2003/04 period, in the relative incidence of ST4 and ST5 (represented by magenta and green bar, respectively), which
account for the 43% of total snowfall events occurring in MVOBS.
At the light of these results, it is reasonable to envisage that the alteration of MVOBS nivometric regime observed among
1970s and 1990s is strongly related to variations in large-scale atmospheric circulation. A comprehensive investigation about
this aspect is offered in the next section.

**4 Relationship with teleconnection indices**
In this section, we investigate connections between ST frequency of occurrence and five teleconnections patterns, introduced
in paragraph 2.5, that have a large influence on Central Mediterranean atmospheric variability.
Firstly, we have performed a simple linear correlation analysis between each ST and AO, NAO, EAWR, EMP and SCAND
indices over the period 1950-2015 (Table 4). In computing the linear correlation, we have considered only the winter seasons
in which the value of a determined index belongs to its upper or lower quartile. This choice allowed better emphasising the
relationship between ST and positive and negative phases of teleconnection patterns (Cohen et al., 2013). Similarly to Table
3, the results are presented in terms of $r$ and significance level: values in bold indicate correlations with 90% significance level.
From a detailed inspection of Table 4, it emerges that $r$ values are generally lower than 0.5, except for ST2, which shows a
very good agreement with NAO and AO indexes ($r = -0.64$ and $-0.70$, respectively); at the same time, it is important
highlighting that all ST are correlated at the 90% significance level with at least one teleconnection index. The ST1 has a low
correlation degree with all indexes, although at 90% confidence level with NAO, AO and EMP. The ST3 is negatively
correlated at 90% significance level with EMP and positively correlated at 90% with NAO and AO indices. ST4 and ST5 are
both positively correlated with EMP ($r = 0.47$ and $0.42$, respectively); moreover, ST4 shows a good agreement with SCAND
index, whereas ST5 is positively correlated with AO index ($r = 0.40$). The last cluster, ST6, exhibits a good linkage only with
EAWR pattern ($r = 0.48$).
It should be noted that we have carried out the correlation analysis also in the scenario in which all teleconnection indices
values are considered. The results are generally similar to the ones obtained considering only the upper and lower quartiles,
although the correlation levels appear to be lower.
Figure 9 provides more compelling evidence about the impact of positive (left panel) and negative (right panel) phase of
teleconnection indices on the ST frequency of occurrence. The latter is expressed in terms of anomaly (number of days per
winter season) with respect to 1981-2010 average. It is worth noting that a determined phase of a teleconnection index may
have very different impacts on the ST regulating the snowfall regime in MVOBS. The positive phase of AO and NAO indexes
determine a positive anomaly of ST3 frequency of occurrence and, conversely, a negative anomaly of ST1, ST2, ST4 and ST6.
The opposite state of the AO pattern (negative phase) is associated, as one would expect, with an increase of ST1 and ST2.

The negative phase of the NAO index seems to exert an appreciable impact only on ST2, whose frequency of occurrence is higher in this circumstance, and on ST3 and ST5, which exhibit an incidence lower than normal. The positive phase of EMP index is clearly linked to a positive anomaly of ST1, ST3, ST4 and ST5 frequency of occurrence and, on the contrary, to a negative anomaly of ST6. The opposite is true for the negative phase. This result can be related to the synoptic scenario synthesised by the positive phase of EMP, characterised by a blocking ridge over north Atlantic that promotes polar or arctic cold air outbreaks across central Europe and Mediterranean regions. The SCAND pattern, in its positive phase, promotes a rising of ST3, ST4, and ST6 incidence, a weak increase of ST2 and ST5 and, at the same time, a decline of ST1 occurrence. The opposite state of this index has a remarkable link only with ST4, whose frequency of occurrence reduces to about 1.5 days per winter season. Regarding EAWR, both its positive and negative phase strongly affect ST2 and ST6, causing a lowering and an increase of the incidence of these patterns, respectively.

According to these findings, it may be helpful to inspect the temporal variability of the teleconnections indices, to search connections with the time behaviour of the six ST identified in this study. In Figure 10 (left panel), we have condensed the wintertime series for the 1950/51-2014/15 period of AO, EAWR, EMP, NAO and SCAND indexes. A detailed analysis of this figure reveals that the considered time interval may be subdivided into three sub-periods: 1950-mid-1970s, in which AO index is generally in its neutral or negative phase and a strong interannual variability prevail for the other patterns (and in particular for NAO index). Subsequently, from the mid-1970s to the end of 1990s, when a strong reduction of winter total HNS in MVOBS was observed (see Fig. 10, right panel), both AO and NAO indices exhibited a positive trend. More specifically, the AO index was, on average, in its positive phase in the mid-1970s, in early 1980s, between the end of 1980s and the beginning of 1990s and at the end of 1990s. In these periods, very strong positive anomalies of the NAO index were also detected. Both patterns switched to the opposite state (i.e. negative) only at the end of 1970s and in the middle of both 1980s and 1990s. From the mid-1980s, the positive phase of EAWR was also observed, especially in the first years of 1990s. The EMP and the SCAND resembles the behaviour taken in the precedent period, showing a strong interannual variability; it is important highlighting that a positive phase of these indices occurred in the mid-1970s (only for SCAND), in the early 1980s and in the first years of 1990s. In the last 15 years of the considered period, the AO index was generally weakly negative or neutral, except for the late 2000s, when a strong oscillation from positive to negative phase was observed. The NAO was generally neutral or positive, aside from the interval between the end of 2000s and the early 2010s, when a strong negative phase prevailed.

It is worth noting that from the mid-1970s there is a near-systematic coupling between most relevant negative and positive AO and NAO events, in contrast with the previous time interval (i.e. 1950-mid 1970s), in which there was a prevalent decoupling. The evidence emerged from this analysis about NAO and AO indices was discussed in detail and related to the global warming phenomenon by Cohen and Barlow (2005). The latter concluded that the pattern and magnitude of the global warming trend in the northern hemisphere over the 1972-2004 period is largely independent of the AO and NAO. In other words, there is a strong divergence between trend in surface air temperature and tendency in AO and NAO indices.

The EAWR is neutral or positive until mid-2000s, and then neutral or negative phases prevailed. The EMP and the SCAND
patterns do not show relevant tendencies: the first one was strongly positive in the mid-2000s and in early 2010s, whereas the
second one was generally neutral and weakly negative, except in the early 2010s, when the positive phase was dominant.
Table 5 provides quantitative evidence of the relationship between teleconnection patterns and winter HNS time series. A
negative correlation has been found for AO, NAO and EAWR indices, whereas a low positive relationship has been detected
for SCAND and EMP. However, a result statistically significant at 90% confidence level has been obtained only for AO index,
which is undoubtedly the most impactful teleconnection pattern on MVOBS nivometric regime. However, from our
investigation it is clear that different ways should be explored to in-depth evaluate the relationship between teleconnection and
the snowfall variability observed in the analysed site. For example, a multiple linear regression analysis (e.g. Cohen et al.,
2013) may be a useful approach. This aspect is behind the scope of this work and it is left for future studies.

**5 Discussion**
The variability over the time of the total winter HNS recorded in MVOBS presents some similarities with evidences provided
by past research carried out in the alpine region. The first common point lies in the strong interannual variability, which many
authors reported in their analysis (e.g. Schöner et al., 2009; Scherrer et al., 2013; Terzago et al., 2013). The patterns emerged
from the behaviour over time of MVOBS signal are generally in line with those identified in some previous studies. In this
respect, it is important highlighting that the strong reduction in snowfall amount and frequency of occurrence occurred in
MVOBS in the 1990s and the subsequent recovery in 2000s have been also observed in Switzerland, France, western Italian
Alps and Austria (e.g. Laternser and Schneebeli, 2003; Micheletti, 2008; Vault and Ciafarra, 2010; Scherrer et al., 2013;
Marcolini et al., 2017; Matiu et al., 2021). However, it should be noted that in most of the investigated alpine sites the decline
in snowfall amount, as well as in NSD, occurred after 1980s, whereas in MVOBS it began in the mid-1970s. Moreover, it is
very interesting highlight that the maximum in snowfall amount found in MVOBS time series in 1900-1920 period (see Table
1) has been also detected in Switzerland by Scherrer et al., 2013.
Unfortunately, aside from the Alpine region, the available literature does not offer many other terms of comparison in the
Italian territory. In this respect, two examples are the long-term nivometric time series collected in Parma and Turin, in the Po
plain. The former has been analysed in Diodato et al. (2018) for the 1777-2018 period. The authors reported a decline in
snowfall frequency of occurrence, mainly in the first half of 19[th] century, as well as in the annual length of the snow season,
attributing these changes to large-scale circulation patterns and in particular to the NAO index. However, no significant trend
has been detected for the amount of fresh snow, according to the available data (1868-2018 period). The Turin snowfall series
has been investigated by Leporati and Mercalli (1993). The latter detected a very strong interannual variability both in terms
of NSD and snowfall amounts, similarly to the results achieved for MVOBS. The main relevant dissimilarity lies in the above-

than-normal snowfall amount measured in 1981-1987 period, which are in contrast with the evidence provided not only by MVOBS, but also by many other Alpine monitoring stations.

The synoptic patterns identified in our work exhibit some analogies, but also some differences, with other synoptic types related to snowfall events in Europe. For example, Merino et al. (2014), using a methodological approach based on a multivariate statistical analysis (including both PCA and CA), found four different synoptic types associated with snowfall events on the northwestern Iberian peninsula. The first one is associated with a maritime arctic air advection over western Mediterranean region, the second one with the advection of polar maritime air masses, the third one with the incoming of polar continental air masses over western Mediterranean area, whereas the fourth-one is characterised by a closed cyclonic circulation over the Iberian Peninsula, which produces strong thermal gradient. The second and the third patterns have several commonalities with the ST6 and ST4 of our works, respectively, both in terms of involved air masses and in the upper level flow. The first and the fourth circulation types discusses in Merino et al. (2014) are unfavourable to snowfall events in the southern Apennines area, although the former traces out a large-scale configuration that promotes the incoming of maritime arctic air mass over Mediterranean basin, by analogy with the ST1 of our study.

Moreover, it is also interesting to compare our results with the study of Esteban et al. (2005), which extracted seven different circulation patterns that explain heavy snow precipitation days in Andorra (Pyrenees). Three of these patterns represent an advection from northwestern of polar maritime air masses which resembles the large-scale flow depicted by ST6, whereas other three types have some common points with ST1, ST2 and ST4, showing scenarios characterised by advection of Atlantic or Mediterranean air masses combined with the outbreak of cold air from northern and eastern Europe. There is only one situation, characterised by a low-pressure area northwestern Spain, which strongly departs from scenarios that trigger snowfall events in Southern Italy Apennines.

According to the results of our analysis, it is very reasonable ascribe the negative snowfall amounts and number of events anomaly observed between 1970s and 1990s to the increase in NAO and AO indices values, which cause a reduction of the occurrence of some synoptic patterns, mainly ST1 and ST2, very favourable to the incoming, in Central Mediterranean area, of cold air masses of maritime (polar or arctic) origin. This achievement is in accordance with the findings of Merino et al. (2014), which attribute the decrease in the number of snow days observed in Castilla y León region (Spain) to the increase in the NAO index during winter months throughout the second half of the 20th century. The impact of NAO and AO anomalies was mitigated by the incidence of ST4 and ST5, which remains quite stable due to the occurrence of some periods characterised by positive values of EMP and SCAND indices. The increase in interannual variability of snowfall events detected in the last two decades, as well as the rise in the average amount, can be attributed to large-scale conditions more beneficial for cold outbreaks in central Mediterranean regions, as well represented by rising in frequency of negative AO patterns and by the occurrence of winter seasons modulated by positive EMP and negative EAWR.

## 6 Conclusions

This work documents the snowfall variability observed from late 19th century to recent years in a remote Apennines site (Montevergine) and proves its strong relationship with a wide range of large-scale atmospheric patterns that govern the winter variability in the Central Mediterranean area.

Using a well-known cluster analysis technique (the k-means) and two meteorological fields (i.e. the 500-hPa geopotential heights and the sea level pressure) provided by the 20CRV3 reanalysis product, we have identified for the 1884/85-2014/15 period six different meteorological regimes. The latter highlights that the linkage between Montevergine's snowfall variability and large-scale atmospheric circulation is very strong but also very complex. Two key-features that bring together the identified synoptic types are (i) the presence of a blocking high-pressure anomaly over north Atlantic and/or Scandinavian Peninsula and (ii) the genesis, in central Mediterranean area, of relevant baroclinic conditions, which constitute an essential ingredient for snow events. As testified by the cluster analysis, different meteorological scenarios, including outbreaks of cold air masses of both maritime and continental origin, can satisfy these conditions. In our opinion, this a first important result, which demonstrates the great relevance and representativeness of the long-term meteorological time series collected in Montevergine, which can be considered an ideal site for the study of climate variability in a "local-to-global framework", because its meteorological regime reflects a wide spectrum of large-scale atmospheric patterns.

The examination of the time variability of total winter snowfall amount and frequency of occurrence has revealed a very strong interannual variability as well as the absence of a relevant trend until the early 1970s. Subsequently, between the 1970s and 1990s, there was a sharp decrease in snowfall amounts and in the number of snowfall events. In the period from early 2000s to mid-2010s, the tendency in snowfall events number and amounts is reversed and a more pronounced interannual variability has been newly observed.

With the aid of a supplementary analysis, we found a linkage between the identified synoptic types and five different teleconnection patterns that drive the European atmospheric variability, i.e. AO, EAWR, EMP, NAO and SCAND. Unfortunately, due to the limited availability of teleconnection indices data, this investigation is confined to the period 1950/51-2014/15. The ST1, ST2 and ST6 patterns are strongly connected, through a negative correlation, with AO, NAO and EAWR indices. The occurrence of the third synoptic type, ST3, is partly related to AO, EMP, NAO and SCAND indices, whereas the ST4 has connections with positive AO and NAO phases and with both positive and negative EMP and SCAND indices. The ST5 has linkages with positive EMP phase and with negative AO, EAWR, EMP and NAO patterns.

The behaviour over time of some teleconnection indices, namely NAO, AO and EAWR, has left a relevant footprint on Montevergine winter snowfall variability. From the mid-1970s to the end of 1990s, these indices exhibited a trend towards positive phase, which determined a lowering of ST1 and ST2 synoptic types occurrence. These two patterns play a relevant role in the Montevergine nivometric regime, as testified by average daily snowfall amounts (11.8 and 12.0 cm, respectively) associated with them. The increase in average snowfall amounts observed in the last 15 years of the analysed period can be explained by the reverse tendency of AO and EAWR indices, which results in a rise of ST4, ST2 and ST6.

To conclude, at the light of this study, the snowfall variability and trend observed in Montevergine in the 1884-2015 period have been largely controlled by large-scale atmospheric variability. In the last past 40 years, when most of the global warming occurred, the investigated nivometric regime has been strongly modulated by AO and NAO patterns, which caused, mainly in the period between mid-1970s and 1990s, a strong reduction of synoptic-scale atmospheric patterns associated to maritime cold air advection.

Additional efforts, left for future studies, should be devoted to the evaluation of the role exerted on the winter snowfall variability by the rising in temperature induced by global warming. In some previous works (e.g. Merino et al., 2014), this aspect is considered to have a strong impact on the snowfall trend and variability, although there is not a clear separation between its contribution and the one provided by large-scale atmospheric circulation patterns. The future perspectives also include the comparison between the evidence provided by the Montevergine snowfall time series with other long-term nivometric data collected in the Apennine environment or in the Alpine context. In this respect, we hope that our study can stimulate new efforts and actions to recover and valorise old snowfall records, especially in Italian territory, which has an inestimable asset of historical meteorological observations. Moreover, we aim to extend the analysis carried out in this work to spring and fall snowfall variability observed in Montevergine, to detect and discuss potential dissonances and similarities with the results obtained for the winter season.

**Appendix A: Data collection and measurements practises**

The Montevergine snowfall data, as well as the other meteorological records, were stored in paper-based registers, containing the daily and sub-daily observations collected for each year (Capozzi et al., 2020). Figure A1 shows an extract of the meteorological register for the data measured in the second decade of January 1946, which includes both daily ($HNS_d$ and $SD_d$) and sub-daily data ($HNS_{sd}$ and $SD_{sd}$). The register is formatted in tables according to the standards suggested by the ICO. The columns storing the $SD_{sd}$ (22-24, starting from the left), $HNS_{sd}$ (25-27) and the $HNS_d$ (28) data are bordered in red for the convenience of the reader. All values are expressed in centimetres (cm).

The standards in terms of measurement instrument and location have not changed over the course of time. The measurement practices frequently changed over time, depending on the standards prescribed by the reference government or military offices, instead. Detailed and useful information about the specific period covered by each parameter involved in nivometric measurements are presented in Fig. A2. More specifically, in some time intervals both daily and sub-daily data are available. Figure A2 shows that near-continuous observations (data availability = 95.7% for the 1884-2020 period) are available only for $HNS_d$ (blue line), which presents a relevant gap in the 1964-1968 period when MVOBS was suspended due to lack of personnel. The $HNS_{sd}$ observations (red line) were carried out only between 1892 and 1963; more specifically, from 1892 to 1932 sub-daily records were collected at 09:00, 15:00 and 21:00 local time, whereas from 1933 to 1963 the three daily data were acquired at 08:00, 14:00 and 19:00 local time. Regarding the SD, this variable was sporadically collected in the periods

November 1944 – March 1952 and November 1953 – May 1961 and then more continuously (although only on a daily basis)
from 1969 to 2008 (black line).

**Code/Data Availability**: The data that support the findings of this study are available from the corresponding author, upon
reasonable request.
**Competing interests**. The authors declare that they have no conflict of interests.
**Author contribution**: Conceptualization, V.C.; methodology, V.C. and C.D.V.; formal analysis, V.C.; investigation, V.C.;
resources, G.B.; writing—original draft preparation, V.C. and C.D.V.; writing—review and editing, G.B., C.D.V. and V.C.;
supervision, G.B. All authors have read and agreed to the published version of the manuscript.
**Acknowledgments**. The authors of this work are very grateful to the Benedectine Community of Montevergine for affording
the opportunity to analyse and digitise the old diaries and meteorological registers stored in Montevergine Abbey. In this
respect, we address special thanks to Reverend Father Abbot Riccardo Guariglia and to Father Benedetto Komar.

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

 **TABLES**


**Table 1**. Average and standard deviation values (in cm) of total winter height of new snow (HNS) and number of snow days (NSD) observed
in MVOBS for different sub-periods.

| Sub-period | Total HNS | Total NSD |
|---|---|---|
| | Average and standard deviation (cm) | Average and standard deviation (number of days) |
| 1884/85-1903/04 | 213.2 ± 104.2 | 18.2 ± 7.4 |
| 1904/05-1923/24 | 221.5 ± 103.6 | 20.3 ± 8.1 |
| 1924/25-1943/44 | 199.7 ± 117.4 | 16.1 ± 7.7 |
| 1944/45-1963/64 | 201.3 ± 140.1 | 16.2 ± 9.1 |
| 1964/65-1983/84 | 160.7 ± 85.3 | 13.7 ± 5.7 |
| 1984/85-2003/04 | 114.1 ± 65.6 | 11.9 ± 5.7 |
| 2004/05-2019/20 | 167.4 ± 109.6 | 13.6 ± 7.1 |





**Table 2.** For each of the six synoptic types, the average frequency of occurrence (in percentage) and the average and standard deviation (in
cm) of daily height of new snow observed in MVOBS are shown. All parameters have been computed over the entire analysed period (1884-
890 2015).

| Synoptic Type | Frequency of occurrence (%) | Average and standard deviation of daily snowfall events (cm) |
|---|---|---|
| ST1 (Arctic Maritime) | 17.7 | 11.8 (± 10.4) |
| ST2 (Central Europe Low) | 12.1 | 12.0 (± 10.3) |
| ST3 (Continental Air) | 14.8 | 9.8 (± 8.6) |
| ST4 (Mediterranean Low) | 19.8 | 12.6 (± 10.8) |
| ST5 (Arctic Trough) | 16.3 | 9.7 (± 7.9) |
| ST6 (Polar Maritime) | 19.3 | 10.2 (± 9.0) |



**Table 3.** Linear correlation coefficient (r) between synoptic types frequency of occurrence and total winter height of new snow (HNS). The results are presented according to seven different sub-periods and to the whole considered time interval. Bold values indicate correlations with 90% significance level.

| Synoptic type | 1884/85-1903/04 | 1904/05-1923/24 | 1924/25-1943/44 | 1944/45-1963/64 | 1964/65-1983/84 | 1984/85-2003/04 | 2004/05-2014/15 | All years |
|---|---|---|---|---|---|---|---|---|
| ST1 | **0.73** | **0.45** | **0.67** | 0.28 | **0.62** | **0.61** | **0.58** | **0.56** |
| ST2 | **0.64** | 0.13 | **0.68** | **0.63** | 0.33 | **0.43** | 0.29 | **0.52** |
| ST3 | -0.05 | **0.41** | **0.40** | **0.46** | 0.10 | 0.05 | **0.62** | **0.27** |
| ST4 | **0.41** | **0.58** | **0.50** | **0.78** | 0.24 | **0.72** | **0.75** | **0.60** |
| ST5 | 0.00 | 0.23 | 0.00 | **0.41** | **0.54** | 0.16 | 0.50 | **0.21** |
| ST6 | 0.36 | **0.44** | 0.38 | **0.41** | 0.11 | 0.34 | 0.10 | **0.36** |

**Table 4.** Linear correlation coefficient (*r*) between synoptic types frequency of occurrence and upper and lower quartiles of teleconnection indices over the period 1950-2015. Bold values indicate correlations with 90% significance level.

| Synoptic type | NAO | AO | SCAND | EAWR | EMP |
|---|---|---|---|---|---|
| ST1 | **-0.30** | **-0.31** | -0.29 | -0.03 | **0.20** |
| ST2 | **-0.64** | **-0.70** | -0.04 | **-0.46** | 0.07 |
| ST3 | **0.33** | **0.38** | 0.22 | 0.13 | **0.37** |
| ST4 | -0.14 | -0.17 | **0.40** | -0.01 | **0.47** |
| ST5 | 0.29 | **0.40** | 0.07 | 0.14 | **0.42** |
| ST6 | -0.34 | -0.29 | 0.18 | **-0.48** | -0.15 |

**Table 5.** Linear correlation coefficient (r) between winter total height of new snow (HNS) and teleconnection indices (all values) over the period 1950-2015. Bold values indicate correlations with 90% significance level.

| Teleconnection index | Linear correlation coefficient |
|---|---|
| AO | **-0.41** |
| EAWR | -0.15 |
| EMP | 0.18 |
| NAO | -0.22 |
| SCAND | 0.25 |

907                                              **FIGURES**

908

909

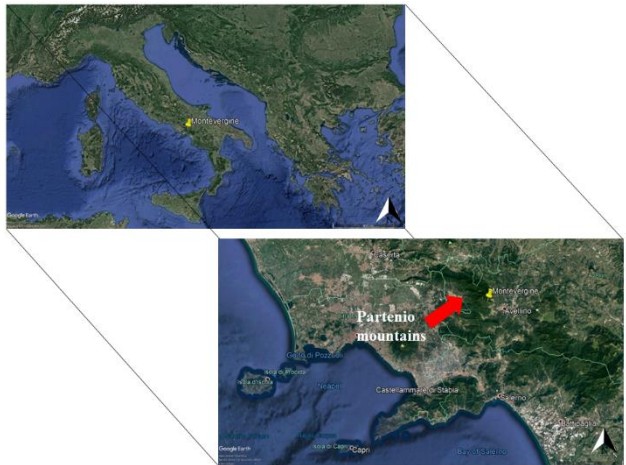 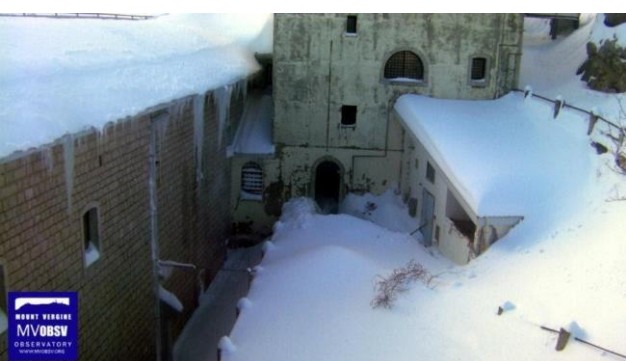

910

**Figure 1: On left panels, the Italian and Balkan peninsulas (in the upper) and the Campania region (in the bottom) are shown. The location of Montevergine observatory is highlighted via a yellow pin, whereas the red arrow indicates the Partenio Mountains. Images credits: © Google Earth, Data Sio, NOAA, U.S. Navy, NGA, GEBCO. The right panel shows a picture (taken on 15 Feb. 2012) of the "*Giardinetto dell'Ave Maria*", a cloister of the Montevergine Abbey where snow measurements have been performed over the time. Image credits: non-profit organization "MVOBSV – Mount Vergine Observatory".**




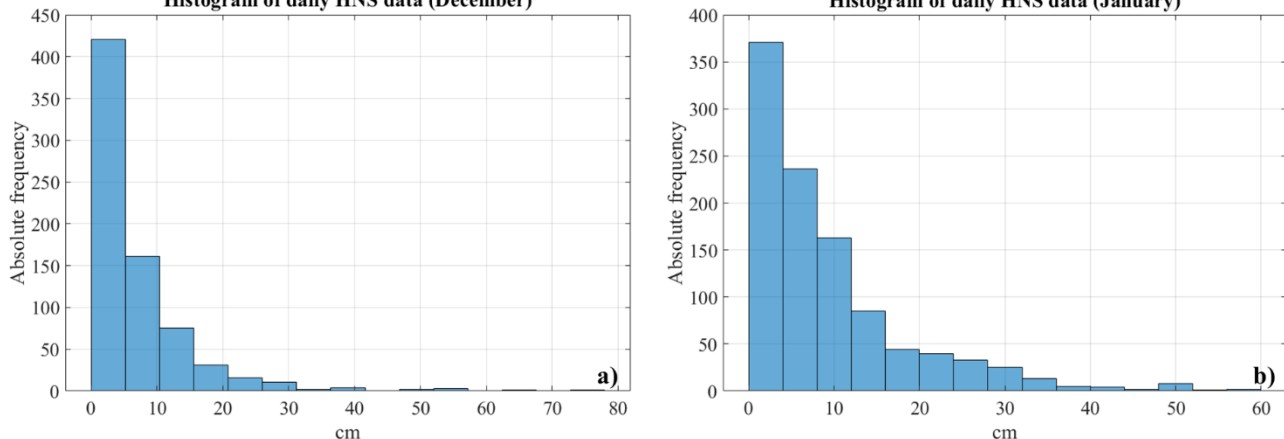


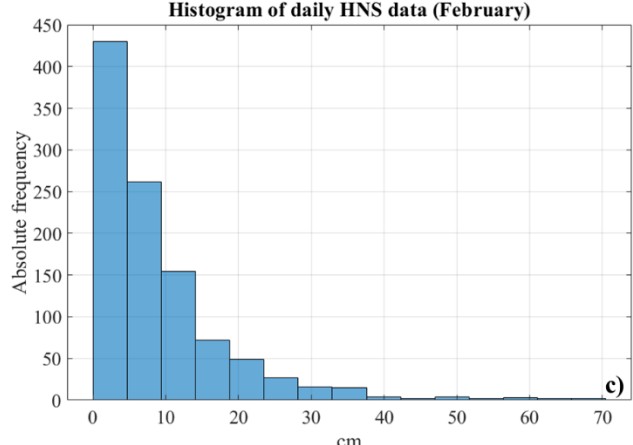


**Figure 2: Histograms of daily height of snow (HNS$_d$) observed in MVOBS in December (a), January (b) and February (c). The y-axis is the absolute frequency (expressed as number of days), whereas the x-axis is the HNS$_d$ amount at the bin centre (in cm). Data collected between 1884 and 2020 have been taken into account.**


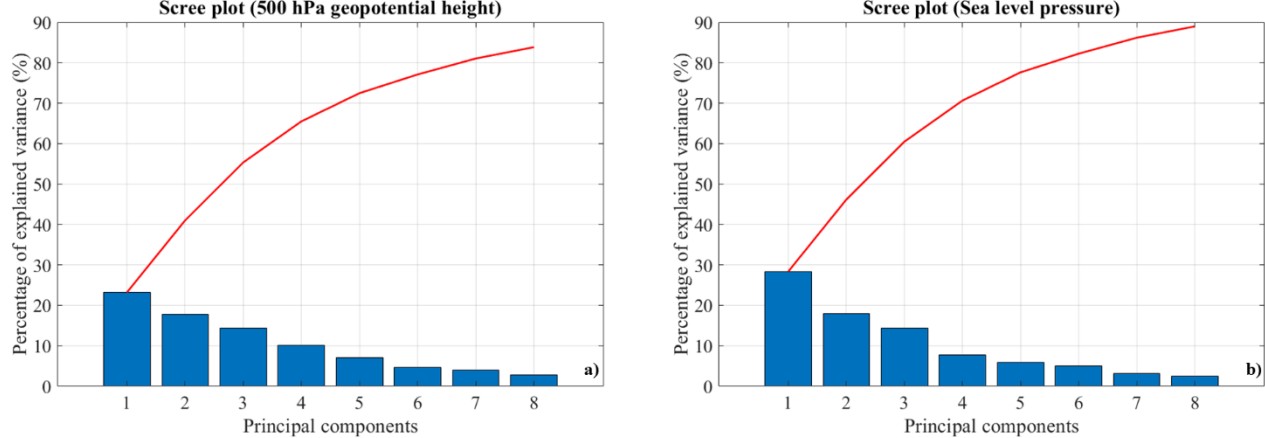


Figure 3: Scree plot of variance (blue bars) and cumulative variance (red line) for the first eight principal components of the two parameters involved in cluster analysis, the 500-hPa geopotential height (a) and the sea level pressure (b). The data have been retrieved from 20CRV3 dataset, considering only winter snow days occurred in MVOBS between 1884 and 2015.



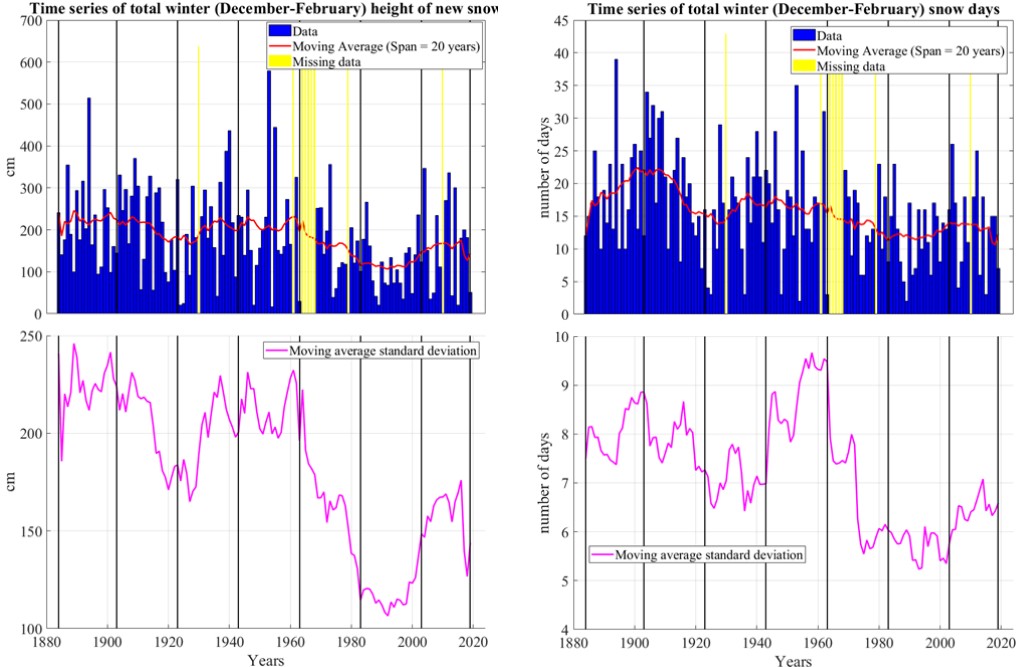


Figure 4: Winter (December to February) time series of total height of new snow (upper left panel) and total number of snow days (upper right panel). In both panels, the missing data are highlighted as yellow bars. The red line shows the 20-years moving average smoothing. The bottom panels present, for total height of new snow (left) and total number of snow days (right), the standard deviation of the moving average (magenta lines). On both panels, the vertical black lines define the limits of the 20-years sub-periods introduced in Table 1.

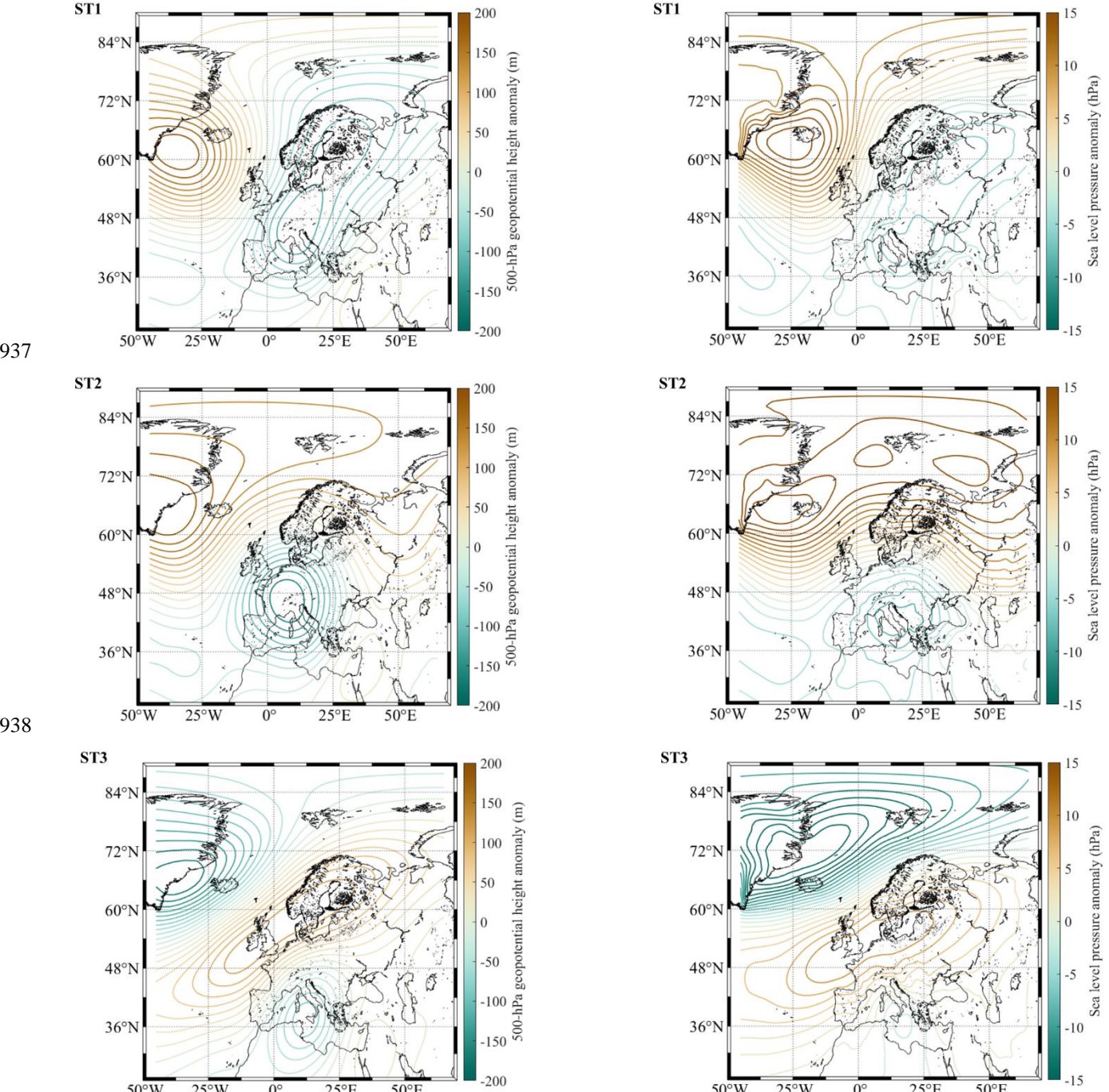




**Figure 5: Synoptic types (ST) controlling the snowfall events and variability in MVOBS. More specifically, this figure sketches the**
**ST1 ("Arctic Maritime"), the ST2 ("Central Europe Low") and the ST3 ("Continental Air"). The left panels (a, c and e) show, for**
**ST1, ST2 and ST3, respectively, the 500-hPa geopotential height anomaly (in m) with a contour interval of 20 m; the right panels (b,**
**d and f) present the sea level pressure anomaly (in hPa) with a contour interval of 1.5 hPa. The ST have been obtained from 20CRV3**
**reanalysis product (1884-2015 period), considering an area embracing the entire European territory (25-90°N, -45-65°E).**

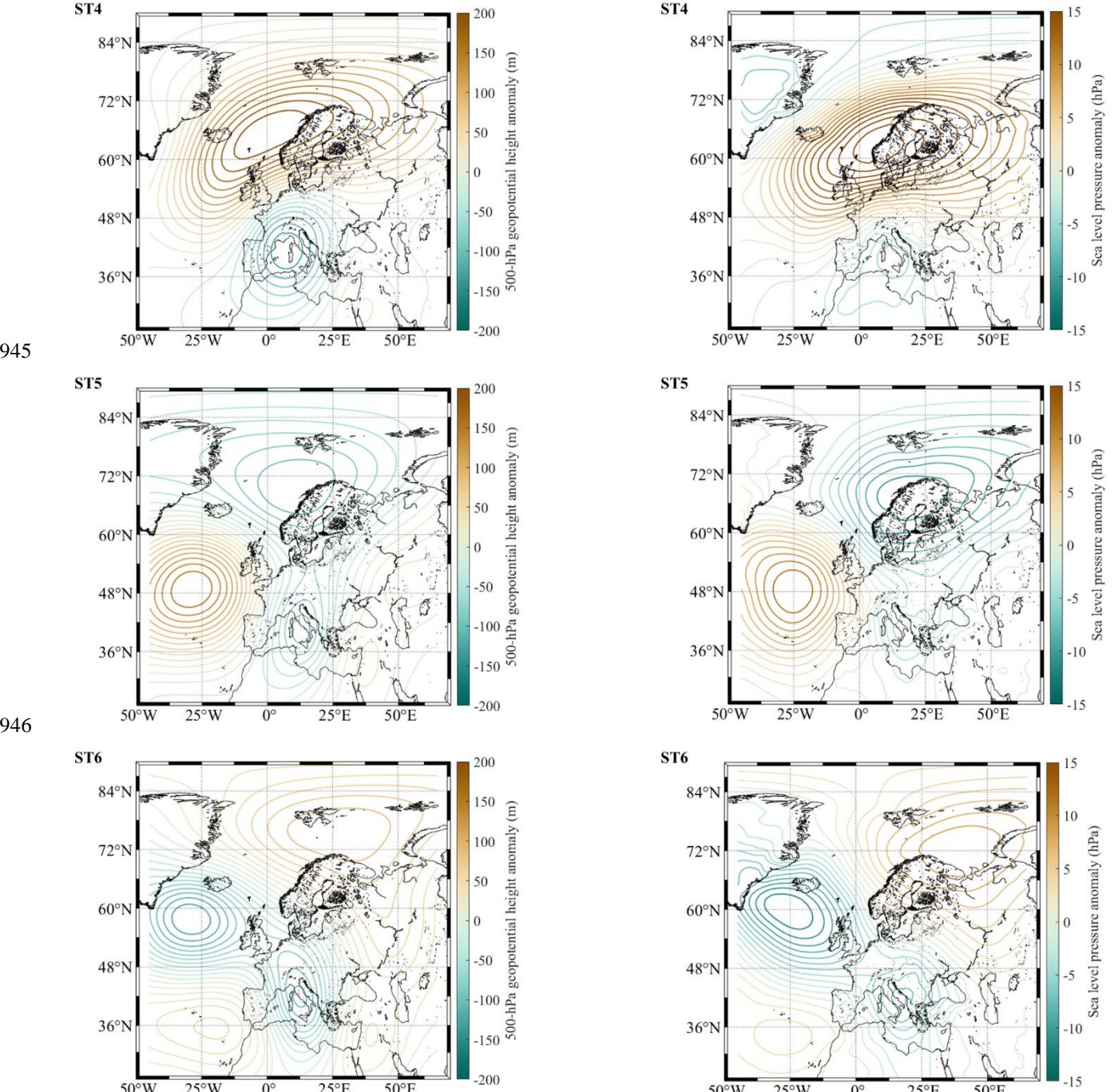

**Figure 6: Synoptic types (ST) controlling the snowfall events and variability in MVOBS. More specifically, this figure sketches the ST4 ("Mediterranean Low"), the ST5 ("Arctic Trough") and the ST6 ("Arctic Maritime"). The left panels (a, c and e) show for ST4, ST5 and ST6, respectively, the 500-hPa geopotential height anomaly (in m) with a contour interval of 20 m; the right panels (b, d and f) present the sea level pressure anomaly (in hPa) with a contour interval of 1.5 hPa. The ST have been obtained from 20CRV3 reanalysis product (1884-2015 period), considering an area embracing the entire European territory (25-90°N, -45-65°E).**

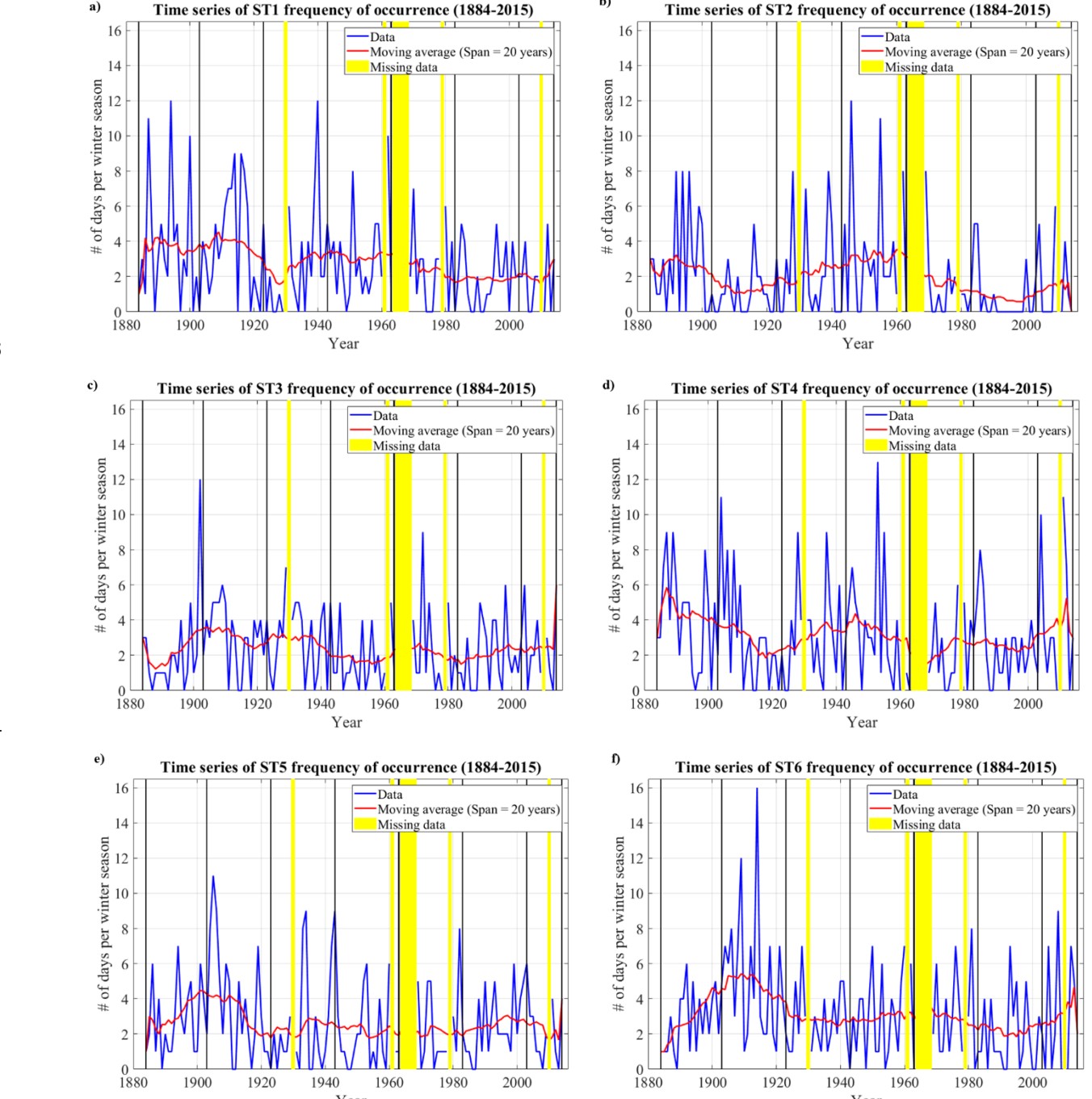

953

954

955

**Figure 7: Time series of the frequency of occurrence, expressed in terms of number of days per winter season, of the six synoptic types (ST) emerged from the cluster analysis, i.e. ST1 (a), ST2 (b), ST3 (c), ST4 (d), ST5 (e) and ST6 (f). In all panels, the missing data are highlighted as yellow bars, whereas the red line shows the 20-year moving average smooth. The period from 1884 to 2015 has been considered.  The red line shows the 20-years moving average smoothing. On all panels, the vertical black lines define the limits of the 20-years sub-periods introduced in Section 3.1, except for the last period, which reduces to 2004/05 to 2014/15.**

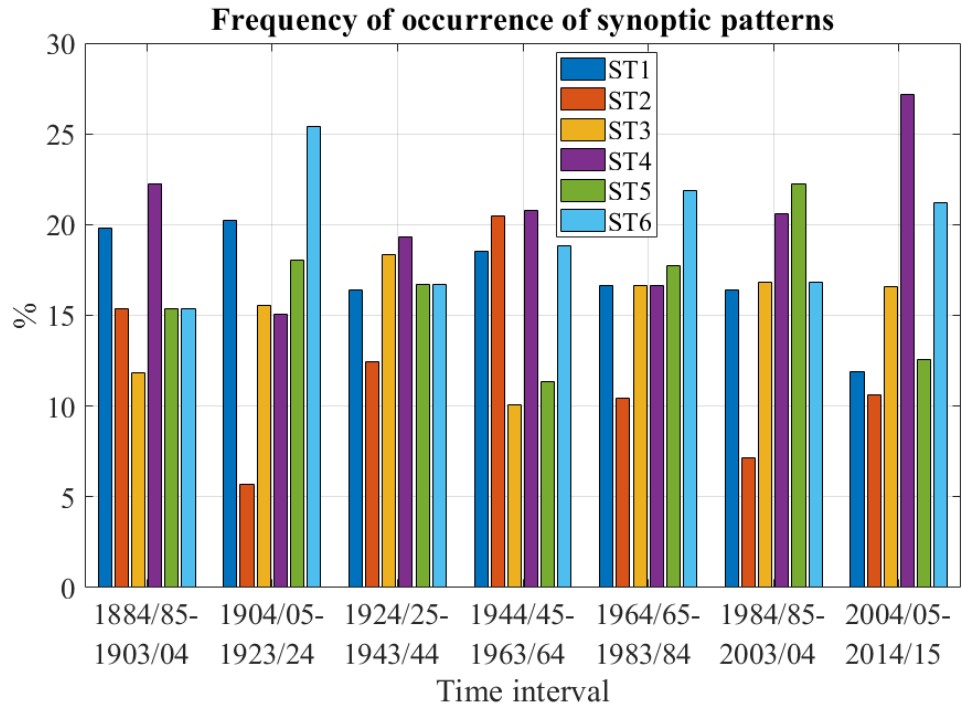

Figure 8: Each group of bars represents the frequency of occurrence of ST (expressed as percentage) in relation to the total number of snowfall events observed in a determined time interval. The six synoptic types, i.e. ST1, ST2, ST3, ST4, ST5 and ST6, are marked as blue, red, orange, magenta, green and cyan bar, respectively.

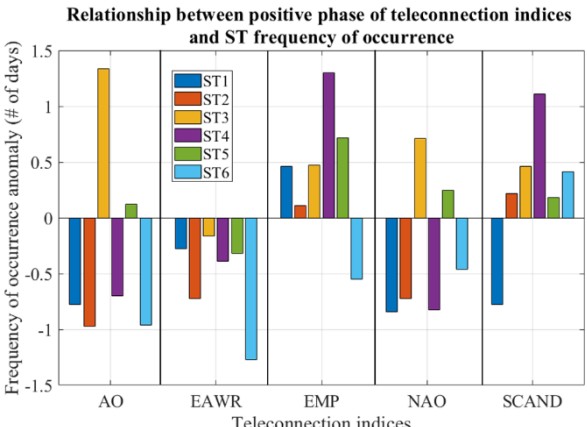
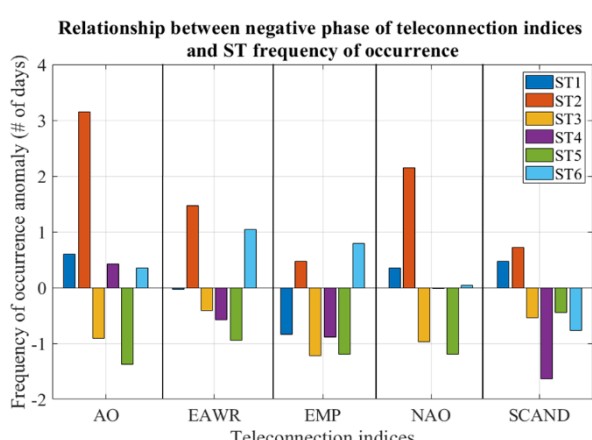

Figure 9: For the positive (left panel) and negative (right panel) phase of the teleconnection indices involved in this study, i.e. the Arctic Oscillation (AO), the Eastern Atlantic Western Russia (EAWR), the Eastern Mediterranean Pattern (EMP), the North Atlantic Oscillation (NAO) and the Scandinavian pattern (SCAND), the frequency of occurrence anomaly of the synoptic types is shown. The anomalies are expressed in number of days and they have been computed with respect to 1981-2010 period. The six synoptic types, i.e. ST1, ST2, ST3, ST4, ST5 and ST6, are marked as blue, red, orange, magenta, green and cyan bars, respectively.

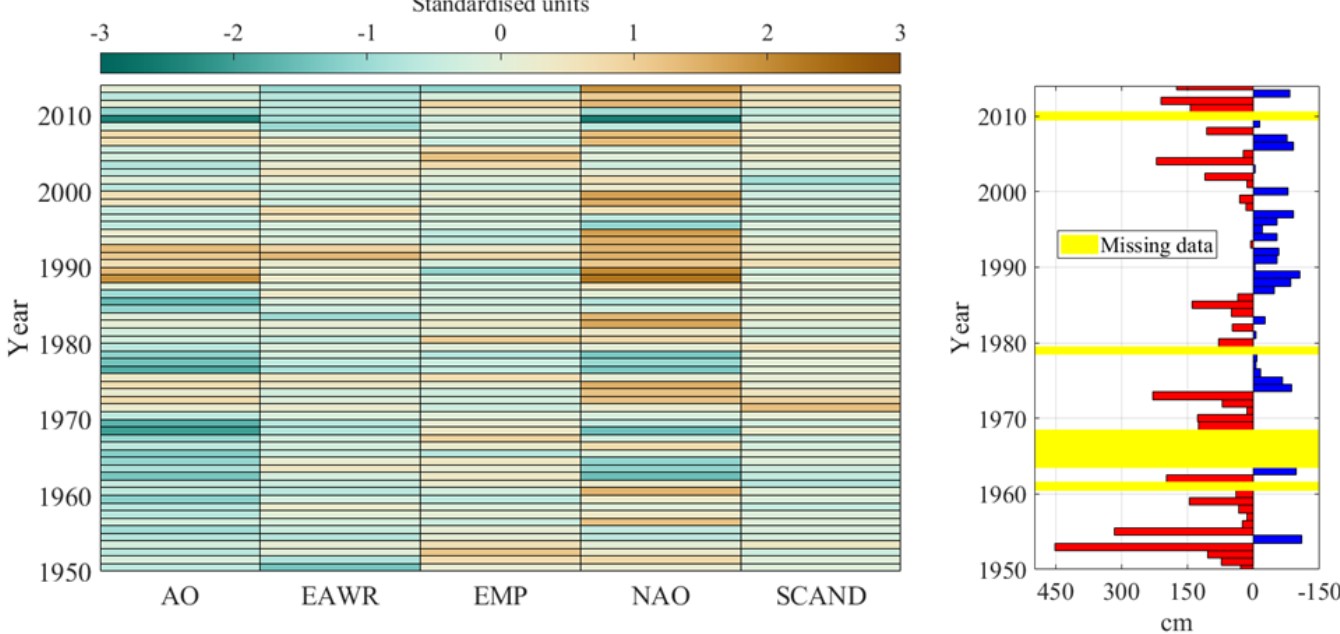

974

**Figure 10: The left panel presents the behaviour in time of the teleconnection indices adopted in this study. More specifically, in the x-axis the winter value of Arctic Oscillation (AO), Eastern Atlantic Western Russia (EAWR), Eastern Mediterranean Pattern (EMP), North Atlantic Oscillation (NAO) and Scandinavia pattern (SCAND) is show from left to right. The indices values are colour-coded according to the horizontal bar and are expressed as standardized units. The right panel shows the winter (December to February) time series of total height of new snow, expressed as anomalies (in cm) with respect to 1981-2010 period. Red bars highlight winters with positive anomaly (in cm), blue bars winters with negative anomaly (in cm). The missing data are marked as yellow bars. For both panels, the period 1950/51-2014/15 is considered.**


Figure A1. This picture presents an example of the original data source (related to the second 10-day period (i.e. from day 11 to 20)
of January 1946). Each row accounts for the observations of a specific day, and each column is devoted to the records of a determined
parameter at a specific hour of the day. The columns including snowfall measurements are bordered with red lines.

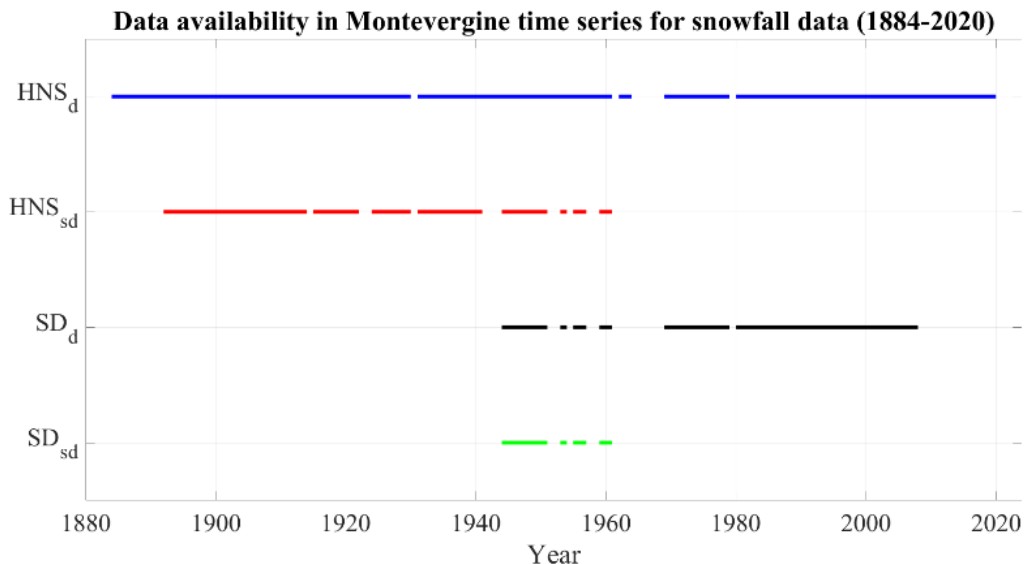

Figure A2: Data availability of snowfall measurements in Montevergine dataset in the period ranging from 1884 to 2020. HNS$_d$ (blue line) stands for daily height of new snow, HNS$_{sd}$ (red line) for sub-daily height of new snow, SD$_d$ (black line) for daily snow depth and SD$_{sd}$ (green line) for sub-daily snow depth parameter. Near-continuous observations are available only for the daily height of new snow (HNS$_d$) parameter.