# Peer review of "Synoptic control over winter snowfall variability observed in a remote"

_The Cryosphere, 2021_

## Author Comment (AC1)

Revision of

**"Synoptic control over winter snowfall variability observed in a remote site of Apennine Mountains (Italy), 1884–2015"**
**V. Capozzi, C. De Vivo, G. Budillon**

**RC** = Referee comment
**AR** = Authors' reply

**REVIEWER #1**
**RC**: Capozzi et al analyze a more than century long series of snowfall from the southern Apennines in Italy, and relate it to synoptic weather types and teleconnection indices. The analysis involves a newly digitised series of snowfall, and adds to the understanding of snowfall in a Mediterranean regime. Since there are little studies on snowfall, especially with such long series and from non-Alpine regions, it is a valuable contribution to the field.

The manuscript is generally well written, even though at times the language is flowery. However, the composition of the manuscript needs to be improved. A lot of methods can be found in the results. The discussion is missing completely (see also point 4 below). And the methods contain too much and too little (point 1 below).

The manuscript focuses a lot on "visual inspection" of time series, as well as relating different time series, but again mostly visual. The manuscript could be significantly improved if the authors would conduct some statistical analyses by relating time series in a bi-variate way, instead of visually comparing time series across pages. This would require only simple correlation analysis, which the authors conducted in other parts of the manuscript. However, extending this type of analysis would make the conclusion and results from the paper much stronger (see also comments below).

**AR:** We are very grateful for his/her positive evaluation of our study and for the time dedicated to the revision of our manuscript. We are also grateful for the comments and the suggestions, which help us to improve our paper and to foster the results.

**Major Comments**

**RC (1):** Methods description Sec 2.1 and 2.2 are both very detailed and at the same time miss critical information. The information on data collection and processing is only interesting for very specific readers. The authors could consider moving large parts of this into an appendix. (also the Petitt and CUSUM are standard tests, so no such detailed description is needed). On the other hand, key information is missing: Which data did you use, daily or subdaily? How did you arrive at monthly values? How did you deal with gaps in the series?

**AR (1):** According to the referee's suggestion, in the revised version of the manuscript a part of section 2.1 will be moved into the Appendix A (Data collection and measurements practises). Moreover, we will delete the details about the Petitt and CUSUM test for data homogenization.
Regarding the reviewer's questions, for our analysis we have used daily height of new snow data (indicated as $HNS_d$ in the manuscript), because they are available for almost all the considered period (1884-2020). The monthly values of snow amount have been computed as a simple sum of all $HNS_d$ data observed in a determined month. About the gaps in the time series, we are not able to reconstruct the missing data due to the unavailability of snowfall time series collected in sites close to Montevergine.

For future works, we are planning to rescue other climatological time series in Campania Region that may include information about snowfall occurrence. Therefore, we will probably able to fill the existing gaps in Montevergine time series only in terms of daily snowfall occurrence, but not in terms of daily snowfall amounts.
We will better clarify these aspects in the revised version of our manuscript.

**RC (2):** L309ff and Table 1: How did you define these periods? Just taking 23yr periods? Why exactly these years? I do not think it's a good idea to create these groups, since they might or might not include and exclude relevant points in the time series. If you want to discuss interannual variability and long-term changes, I suggest employing moving window averages (for long-term changes) and moving window standard deviations (for interannual variability). A period of 20 or 30 years would make sense. This would not have the "problem" of arbitrarily defining year groups.

**AR (2):** Following the referee comment, we have revised the Figure 5 of our manuscript. Note that this Figure is now labelled as Figure 4 after the revision process. More specifically, in order to highlight the long-term changes, we have used the moving window average (with a time span of 20 years), whereas to better emphasize the interannual variability we have computed (and plotted) the moving window standard deviation.

[Figure]

**Figure 4: Winter (December to February) time series of total height of new snow (upper left panel) and total number of snow days (upper right panel). In both panels, the missing data are highlighted as yellow bars. The red line shows the 20-years moving average smoothing. The bottom panels present, for total height of new snow (left) and total number of snow days (right), the standard deviation of the moving average (magenta lines). On both panels, the vertical black lines define the limits of the 20-years sub-periods introduced in Table 1.**

The subdivision of the investigated time interval into sub-periods of 23 years allowed us to emphasize the strong reduction in snowfall amount observed in the period from mid-1970s to the end of 1990s. According to the reviewer's suggestion, we have segmented the time series into more customary 20-years intervals. It should be pointed out that this choice reduces the last sub-period (2004/05-2019/20) to a length of 16 years. We have revised the Table 1 of our manuscript as follows.

**Table 1**. Average and standard deviation values (in cm) of total winter HNS and number of snow days (NSD) observed in MVOBS for different sub-periods.

| Sub-period | Total HNS | NSD |
|---|---|---|
| | Average and standard deviation (cm) | Average and standard deviation (number of days) |
| 1884/85-1903/04 | 213.2 ±104.2 | 18.2 ±7.4 |
| 1904/05-1923/24 | 221.5 ±103.6 | 20.3 ±8.1 |
| 1924/25-1943/44 | 199.7 ±117.4 | 16.1 ±7.7 |
| 1944/45-1963/64 | 201.3 ±140.1 | 16.2 ±9.1 |
| 1964/65-1983/84 | 160.7 ±85.3 | 13.7 ±5.7 |
| 1984/85-2003/04 | 114.1 ±65.6 | 11.9 ±5.7 |
| 2004/05-2019/20 | 167.4 ±109.6 | 13.6±7.1 |

Moreover, we also modified the Figure 9 of our manuscript, which is now labelled as Figure 8.

[Figure]

**Figure 8: Each group of bar represent the frequency of occurrence of ST (expressed as percentage) in relation to the total number of snowfall events observed in a determined time interval. The six synoptic types, i.e. ST1, ST2, ST3, ST4, ST5 and ST6, are marked as blue, red, orange, magenta, green and cyan bar, respectively.**

All these changes will be included in the revised version of the manuscript.

**RC (3):** Figure 8 and related text: Besides the issue with year groups (see comment above), I think it would be much easier if you showed scatter plots of STx versus HNS, instead of trying to compare time series across pages. You could also calculate correlations between these two to give more weight to what you identify from "visual inspection". This would make it easier for readers to see your points.

**AR (3):** We agree with the reviewer remarks. However, the scatter plot between STx and HNS is not a good solution to show the correlation degree between the two variables. The frequency of occurrence of ST, in fact, has a behaviour very similar to a categorical variable. This causes overlapping problems, so in some scatter diagram there are tenths of values all stacked on top of each other. For this reason, we have decided not to modify the Figure 8 of our manuscript (which is now labelled as Figure 7). According to the reviewer's suggestion, we have computed the linear correlation coefficient for each time periods presented in the new version of Table 1 (except for the last period, which reduces to 2004/05 to 2014/15). The results are presented in a following Table, which will be numbered as Table 3 in the revised manuscript.

**Table 3**. Linear correlation coefficient ($r$) between synoptic types frequency of occurrence and total winter height of new snow (HNS). The results are presented according to seven different sub-periods. Bold and grey values indicate correlations with 95% and 90% significance levels, respectively.

| Synoptic type | 1884/85-1903/04 | 1904/05-1923/24 | 1924/25-1943/44 | 1944/45-1963/64 | 1964/65-1983/84 | 1984/85-2003/04 | 2004/05-2014/15 |
|---|---|---|---|---|---|---|---|
| ST1 | **0.73** | **0.45** | **0.67** | 0.28 | **0.62** | **0.61** | 0.58 |
| ST2 | **0.64** | 0.13 | **0.68** | **0.63** | 0.33 | 0.43 | 0.29 |
| ST3 | -0.05 | 0.41 | 0.40 | **0.46** | 0.10 | 0.05 | 0.62 |
| ST4 | 0.41 | **0.58** | **0.50** | **0.78** | 0.24 | **0.72** | **0.75** |
| ST5 | 0.00 | 0.23 | 0.00 | 0.41 | **0.54** | 0.16 | 0.50 |
| ST6 | 0.36 | 0.44 | 0.38 | 0.41 | 0.11 | 0.34 | 0.10 |

Note that the significance of the correlation coefficient has been tested through the well-known p-value method. In the revised version of the manuscript, we will clarify this aspect and we will discuss the results of the correlation analysis presented in the Table 3.

**RC (4):** Discussion is missing completely. Regarding the snowfall series: How do your results compare to other long-term series from Italy, such as Parma or Torino? If I remember correctly they have been published, but possibly not in international journals.

**AR (4):** In the revised version of the manuscript, we will add the following Discussion section, in which we compare our results with Parma and Torino time series. Thank you for the suggestions.

[revised manuscript text omitted]

**Minor Comments**

**RC (1):** L11: mismatch of period wrt to title. OK, later I understood. The snowfall series ends 2020, but the reanalysis 2015, right? You should clarify this and be clearer.

**AR (1):** Yes, it is right. In the revised version of the manuscript, we will specify that the cluster analysis has been applied to 1884-2015 according to the availability of reanalysis data.

**RC (2):** L37ff the literature review is a bit random. It's mixing snow cover parameters (depth, fresh snow, SCD) and it's not clear why the authors chose the specific geographic limit. Btw, there are many other studies from Italy and other countries in the Alps. Also more with century long series. It's not necessary to mention all, but maybe the authors could make their point better.

**AR (2):** Following the referee's suggestion, we have broaden our literature review as follows, including several other references, mainly related to the alpine region.

"The study period of the climatological researches carried out in such works is generally limited to the last 55-65 years. However, few studies (mainly focused on the Alpine area) extended their analysis further back, due to the absence, in many areas, of reliable and continuous old snowfall climatological records. In this respect, it is deserving of mention the study of Scherrer et al. (2013), which investigated the snowfall variability observed in Switzerland during the 1864-2009 period, using nine different stations. According to the findings of this work, the analysed depth of snowfall time series exhibit a strong decadal variability. The highest value in depth of snowfall and days with snowfall occurred in 1900-1920 and 1960-1980 period, the lowest between 1980s and 1900s, whereas an increase in depth of snowfall has been observed in 2000s. Another important reference for mountainous areas is Laternser and Schneebeli (2003), who discovered, for the Swiss Alps, an increase in snow cover and duration from 1930 to early 1980 and, subsequently, a statistically significant decrease towards 1999. Some years later, Beniston (2012b) observed a decline in winter snow depth in the 1930-2010 period by analysing 10 time series in Switzerland. The study of Marty and Blanchet (2012), which has investigated a very large number of snowfall records collected in Switzerland during 1931-2010 period, reached similar conclusions. The authors discovered that the 44% of the considered stations show a significant decrease in the annual snow depth maxima values. Another interesting evidence can be found in the study of Schöner et al. (2009), which examined the snow depth time series collected in Sonnblick (Austria) in the 1928-2005 period: the authors reported a strong interannual variability in snow depth (with largest values in 1940-1950 period) and a decreasing trend in summer snow. Terzago et al. (2013) found a decrease in snow depth (2-14 cm per decade) in the western Italy for the period 1926-2010. Moreover, they also

reported a very strong interannual variability, with maxima in 1940, 1950 and 1960, minima in 1990 and then a recovery in the 2000s. Irannezhad et al. (2017) evaluated the annual snowfall variability in Finland in the 1909-2008 period, analysing its relationship with some large-scale atmospheric patterns. From this study, it emerges a significant decline in annual snowfall in the northern part of Finland, as well as a strong relationship between the observed snowfall variability and the large-scale teleconnection patterns synthesised by Arctic Oscillation and East Atlantic indices."

**RC (3):** Introduction: Long series are great, but snowfall has extremely high spatial variability, so many short series can identify different aspects than one long time series. Maybe the authors could elaborate more on this topic.

**AR (3)**: According to the referee suggestion, we have further broaden the literature review as follows, including more references about snowfall variability observed in the recent 10-years periods.

"Focusing on the Alpine region, Micheletti (2008) carried out an analysis for the Friuli Venezia Giulia (northeast of Italy) for the 1972-2007 period, by investigating the seasonal depth of snowfall records from eight stations. A positive anomaly was observed until the end of 1980s, followed by a lowering of snowfall amount between 1990 and 2000 and by a subsequent recovery (but below the level of 1980s). Valt and Ciafarra (2010) investigated the accumulated snowfall variability observed in Italian Alps (eastern and western sectors) in the period from 1960 to 2009. As general results, they found a negative trend, which is strongest in spring season and below 1500 m asl. Terzago et al. (2010) have provided a focus on the Piedmont Region (northwest of Italy), examining the monthly snow depth and depth of snowfall for the period 1971-2009; the authors discovered an increasing trend for November-December period and a decreasing tendency in January-April. Kreyling and Henry (2011) analysed 177 stations in Germany for the period 1950-2000, focusing on snow days variability: they found a negative trend for the majority of the stations, especially in the last 15 years of the considered period. Marcolini et al. (2017) investigated the snow depth time series collected in the Adige basin (North East Italy) for the period 1980-2019. According to the results of this study, a relevant reduction of both snow cover duration and mean seasonal snow depth occurred in the Adige catchment after 1988, both at low and high elevation sites. On the contrary, in the period from 2000 to 2009, an increase (although not identified as statistically significant) of mean seasonal snow depth has been recorded. Lejeune et al. (2019) found a relevant decrease in the snow depth observed in Col de Porte (France) during the period 1960-2017. Moreover, it is worth mentioning the study of Schöner et al. (2019), which examined a very high number (139) stations located in Austria and Switzerland for the period 1961-2012. The authors of this work found negative tendencies in snow depth (up to -12 cm / 10 years period for sites at elevations of about 2000 m asl) and a positive relationship between the strength of snow depth trends and the altitudes."

**RC (4):** L64ff: At this point in a paper, there should not be a summary of what is being done, but the aims of the paper (high-level understanding, hypothesis, etc.).

**AR (4)**: According to the referee remarks, we have reformulated this part of the Introduction section as follows.
"The main purposes of our work can be synthetized as follows:

- To extend, from both quantitative and qualitative perspectives, the current knowledge about the past snowfall variability observed in Mediterranean mountainous sectors;
- To shed light on links between large-scale atmospheric circulation and local climate variability, by identifying and analysing the synoptic patterns favourable to winter snowfall events in Montevergine as well as their relationship with the main teleconnection indices that govern the atmospheric circulation in the Mediterranean area."

**RC (5):** Figure 5: why did you choose a lowess smoother? Would a simple moving average (10/20/or 30 years) be easier? For the lowess, you also need to supply the degree and the weights, not only the time span.

**AR (5)**: We have chosen the lowess smoothing because it generally works better than moving average at the edges of the time series. However, following the referee suggestion, we have replaced the lowess smoothing with the moving average (See the Figure 4 in the previous comment).

**RC (6):** How are "snow days" (NSD) defined? This would belong in the methods. (related Table 1, Figure 5, …)

**AR (6)**: Usually, a "snow day" is a day on which accumulated snowfall (i.e. daily high of new snow, $HNS_d$) is at least 1.0 cm. However, in our work, in applying the cluster analysis (CA) we have used a slight different definition of "snow day". More specifically, we have considered as "snowy" a day in which the recorded $HNS_d$ value was at least 3.0 cm. This threshold allows filtering out most of some ambiguous events, characterized by the simultaneous presence of different hydrometeors types (i.e. rain, snow hail or graupel).
In the revised version of our manuscript, we will better clarified this point.

**RC (7):** L409: how did you determine statistical significance of trends?

**AR (7)**: Sorry for missing this important detail. To compute the statistical significance of trend, we used the Mann-Kendall test (Mann 1945, Kendall 1962). We will clarify this aspect in the revised version of the manuscript.

**RC (8):** L452: How does the correlation table look like for all values, not only the upper and lower quartiles? Maybe for the supplement.

**AR (8)**: In the following Table, we present the correlation analysis in the scenario in which all teleconnection indices values are considered. The results are generally similar to the ones obtained considering only the upper and lower quartiles, although the correlation levels appear to be generally lower. In the revised manuscript, we added a brief sentence about this aspect, but we decide to not include this Table.

Linear correlation coefficient (r) between synoptic types frequency of occurrence and teleconnection indices (all values) over the period 1950-2015. Bold and grey values indicate correlations with 95% and 90% significance levels, respectively.

| Synoptic type | NAO | AO | SCAND | EAWR | EMP |
|---------------|------|------|-------|-------|------|
| ST1 | -0.16 | -0.19 | -0.21 | -0.05 | 0.14 |
| ST2 | **-0.49** | **-0.59** | -0.06 | **-0.31** | 0.05 |
| ST3 | **0.27** | **0.36** | 0.17 | 0.07 | **0.23** |
| ST4 | -0.05 | -0.04 | **0.27** | -0.06 | **0.35** |
| ST5 | **0.30** | 0.21 | 0.06 | 0.12 | **0.32** |
| ST6 | -0.08 | -0.13 | 0.10 | **-0.39** | -0.12 |

**RC (9):** Section 4: have you considered also correlating teleconnection indices to the HNS series?

**AR (9)**: Our main aim is to search for relationship between the identified synoptic patterns and the teleconnection indices. However, we have welcomed the suggestion of the referee and we have computed the linear correlation coefficient between the winter HNS time series and the teleconnection indices. As demonstrated by the following table, the correlations are generally low, except for AO index, which is positively correlated to HNS time series at 95% significance level. We will include this Table in the revised version of the manuscript (Table 5).

**Table 5.** Linear correlation coefficient (r) between winter total height of new snow and teleconnection indices (all values) over the period 1950-2015. Bold values indicate correlations with 95% significance level.

| Teleconnection index | Linear correlation coefficient |
|----------------------|-------------------------------|
| AO | **-0.41** |
| EAWR | -0.15 |
| EMP | 0.18 |
| NAO | -0.22 |
| SCAND | 0.25 |

**RC (10):** Figure 11: Hovmöller plots do not work for a discrete x-axis, where you have the five teleconnection indices. Would a simple correlation analysis not work better here, too?

**AR (10)**: We understand the doubts of the reviewer with respect to the Figure 11 of our manuscript (which is now labelled as Fig. 10). However, we feel that this picture give a complete, comprehensive and simple, even though qualitative, representation of the relationship between the analysed teleconnection indices and the HNS time series. In our opinion, a linear correlation analysis, in which the indices are considered separately from each other, does not say much about the linkages with the nivometric regime of the site of interest.
A possible approach to investigate about the relative influences of the indices on the winter HNS may be a multiple linear regression analysis (e.g. Cohen et al., 2013). However, a quantitative and in-depth evaluation of this aspect is left for a future work. Therefore, we decide to leave unchanged the Figure 10 in the revised version of the manuscript and to include the linear correlation analysis suggested by the reviewer (see the previous comment).

**List of references**

[revised manuscript text omitted]

---

## Author Comment (AC2)

Revision of

**"Synoptic control over winter snowfall variability observed in a remote site of Apennine Mountains (Italy), 1884–2015"**
*V. Capozzi, C. De Vivo, G. Budillon*

**RC** = Referee comment
**AR** = Authors' reply

**REVIEWER #2**

**RC:** I appreciate the opportunity to review this very interesting and generally well-written work related to snowfall in Italy. This study uses a unique snowfall record from the Montevergine Observatory in Italy's
Apennine Mountains to investigate snowfall variability over the winter months (DJF) between 1884 and 2015. Via cluster analysis, the authors identify six synoptic atmospheric circulation patterns conducive to snowfall in the Apennine Mountains and link observed snowfall variability in their time series to changing frequencies in these synoptic types. Finally, the authors analyze the relationship between the synoptic types identified in this work and five teleconnection patterns important for winter weather in Europe. The findings from this study indicate synoptic-scale atmospheric variability largely controls snowfall variability at the Montevergine Observatory. This work falls squarely within TC's scope. The unique snowfall time series presented here provides snow information from a lesser-studied region and is particularly noteworthy and interesting for its broad temporal coverage and daily resolution. By combining this record with analyses of synoptic-scale atmospheric circulation, this study contributes valuable climatic information and knowledge about the mountain cryosphere which will be of interest to a broad audience. I found the manuscript enjoyable to read and generally well-written. Grammatical errors do occasionally hinder understanding or serve as a distraction to the authors' overall message (see examples under Technical corrections) – however, I believe the authors can quickly correct most of these errors. Similarly, figures are relevant and mostly readable, although I have made some suggestions for improvements in the specific comments below. The organization of the manuscript made sense to me, but I think a discussion of the results should be more explicitly highlighted either via the addition of a Discussion section or via individual discussion subsections throughout Sections 3 and 4.

**AR:** Dear Dr. Hancock, we are very grateful for your positive comments about our study. We are very grateful for all suggestions and remarks, which contributed to improve our manuscript in a substantial way. Thank you very much for the time dedicated to the revision of our work.

In the revised version of the manuscript, we will include the following Discussion section, in which we compare our findings with the results of previous studies.

[revised manuscript text omitted]

**RC**: My main concerns with the manuscript relate to the analyses of the snowfall time series (Section 3.1) and the section analyzing synoptic type variability in time (Section 3.3). In these sections, the combination of the subjective sub-period time interval selection and the visual time series inspections impede result robustness. A more objective method for analyzing variability in the time series – I like the other review's moving window suggestion – would mostly resolve my concerns in these sections by eliminating the subjectivity in the sub-period selection. If the authors wish to continue using the subperiods as presented in the current manuscript, the rationale behind the sub-period selection should be specifically addressed in Section 2 (Materials and methods). In this case, employing some statistical time series analyses in addition to the visual time series inspections would be necessary.

**AR:** The subdivision of the investigated time interval into sub-periods of 23 years allowed us to emphasize the strong reduction in snowfall amount observed in the period from mid-1970s to the end of 1990s. According to this suggestion and to remarks of the referee #1, we have segmented the time series into a more standard 20-years intervals. It should be pointed out that this choice reduces the last sub-period (2004/05-2019/20) to a length of 16 years (see Table 1 of the revised manuscript). Moreover, we have applied the moving average smoothing to both HNS and NSD time series, using a window of 20 years. To better emphasize the interannual variability we have computed (and plotted) the moving window standard deviation (see the following Figure, which will be labelled as Figure 4 in the revised version of the manuscript).

[Figure]

**Figure 4: Winter (December to February) time series of total height of new snow (upper left panel) and total number of snow days (upper right panel). In both panels, the missing data are highlighted as yellow bars. The red line shows the 20-years moving average**

**smoothing. The bottom panels present, for total height of new snow (left) and total number of snow days (right), the standard deviation of the moving average (magenta lines). On both panels, the vertical black lines define the limits of the 20-years sub-periods introduced in Table 1.**

**Specific comments:**

**RC (1)**: Line 160 – On Figure 3, it appears several HNSd values exceeded the 35 cm threshold and would actually have been detected by the gap check? Am I misinterpreting what you have done here in terms of identifying outliers?

**AR (1):** In the framework of quality control, the gap check has been applied as follows. Firstly, for each month we have sorted in ascend order the $HNS_d$ data recorded over the entire available period (1884-2020). Subsequently, we have calculated the difference between two-consecutive values: if a certain $HNS_d$ value is at least 35 cm larger than the previous record, then that value and subsequent ones are flagged as outliers. Therefore, in the Figure 3 (which will be labelled as Figure 2 in the revised manuscript), the $HNS_d$ values exceeding the 35 cm threshold should not to be interpreted as outliers. This figure simply present the frequency distribution of $HNS_d$ data for the three investigated winter months (December, January and February) and it demonstrates that there are not cases in which the difference between two consecutive bins exceeds the selected threshold (35 cm).

**RC (2)**: Lines 226-227 – a sentence or reference justifying the selection of your domain would be helpful here.

**AR (2):** In the revised manuscript, we will add the following sentence: **"**The dimensions of the domain were selected to be consistent with the synoptic scale analysis of the present work and to avoid circulation features in regions remote from the study area (Merino et al., 2014)".

**RC (3)**: Line 237 – here the HNSd threshold to determine NSD is 3 cm, but in Lines 331-332 related to Figure 5 you state that the right panel of Figure 5 includes days with over 1 cm of snow. Why do you display different data in Figure 5 than you use to determine the snow days for synoptic typing? Since Figure 5 is the only location where the total NSD is displayed as a time series (e.g. Figure 8 does not include a panel showing aggregate snow days from all synoptic types), I think it is important the NSD in Figure 5 match the 1986 days used in the cluster analyses.

**AR (3):** Ok, we have modified the right panel of the Figure according to the referee suggestion (see the Figure 4 in the previous comment). Therefore, in the revised version of the manuscript, the total winter NSD will be computed by considering a "snow day" as a day on which accumulated snowfall is at least 3.0 cm.

**RC (4)**: Lines 336-340 – I appreciate the plain-language labelling of the synoptic types here, but these names are not used consistently throughout the remainder of the work. I'd recommend sticking with just

one of the naming conventions, e.g. either ST1 or Arctic Maritime, or including the plain language name parenthetically as in Table 2.

AR (4): Ok, we have adopted the "ST" convention to label the synoptic patterns. Therefore, we have removed the labels "Arctic Maritime", "Central Europe Low", "Continental Air", "Mediterranean Low", "Arctic Trough" and "Polar Maritime".

RC (5): Section 3.2 – This section would really benefit from discussion comparing the synoptic types you have identified with other synoptic work related to snowfall in Europe. I realize you only included snow days in your analyses, but I am also curious about the prevailing synoptic conditions which do not result in snowfall in the Apennines. Even just a couple sentences about this in a discussion would be helpful.

AR (5): In the revised version of our paper, we will discuss this point as follows:

"The synoptic patterns identified in our work exhibit some analogies, but also some differences, with other synoptic types related to snowfall events in Europe. For example, Merino et al. (2014), using a methodological approach based on a multivariate statistical analysis (including both PCA and CA), found four different synoptic types associated with snowfall events on the northwestern Iberian peninsula. The first one is associated with a maritime arctic air advection over western Mediterranean region, the second one with the advection of polar maritime air masses, the third one with the incoming of polar continental air masses over western Mediterranean area, whereas the fourth-one is characterised by a closed cyclonic circulation over the Iberian Peninsula, which produces strong thermal gradient. The second and the third patterns have several commonalities with the ST6 and ST4 of our works, respectively, both in terms of involved air masses and in the upper level flow. The first and the fourth circulation types discusses in Merino et al. (2014) are unfavourable to snowfall events in the southern Apennines area, although the former traces out a large-scale configuration that promotes the incoming of maritime arctic air mass over Mediterranean basin, by analogy with the ST1 of our study.
Moreover, it is also interesting to compare our results with the study of Esteban et al. (2005), which extracted seven different circulation patterns that explain heavy snow precipitation days in Andorra (Pyrenees). Three of these patterns represent an advection from northwestern of polar maritime air masses which resembles the large-scale flow depicted by ST6, whereas other three types have some common points with ST1, ST2 and ST4, showing scenarios characterised by advection of Atlantic or Mediterranean air masses combined with the outbreak of cold air from northern and eastern Europe. There is only one situation, characterised by a low-pressure area northwestern Spain, which strongly departs from scenarios that trigger snowfall events in Southern Italy Apennines."

RC (6): Lines 397-400 – Are these differences statistically significant?

AR (6): We have evaluated the significance of these differences using the popular t-test method. The latter test the hypothesis that two independent samples come from distributions with equal means. More specifically, we have tested the significance between the average $HSN_d$ found for ST1, ST2 and ST4 (11.8, 12.0 and 12.6 cm, respectively) and the average $HSN_d$ values found for ST3, ST5 and ST6 (9.9, 9.7 and 10.2 cm, respectively). According to the p-value, (i.e. the probability of observing the given result, or one more extreme, by chance if the null hypothesis is true), the null hypothesis (i.e. the average

HSNd values are equal) can be rejected at the 5% level for all ST couples (for examples, ST1 vs ST5, ST2 vs ST6 and ST4 vs ST3). In the revised version of the manuscript, we will better clarify this point.

**RC (7)**: Figure 5 – See the comments above related the number of snow days in the right panel. If you elect to continue with the sub-periods, please delineate the time periods in the graph to help the reader.

**AR (7):** Ok, we have modified the Figure 5 (which is now labelled as Figure 4) according to the referee's comment.

**RC (8)**: Figure 8 – If you elect to continue with the sub-periods, please delineate the time periods in the graph to help the reader. Please also consider including a panel showing the total NSD, if the data in Figure 5 are different.

**AR (8):** Ok, we have modified the Figure 8 (which is now labelled as Figure 7) according to the referee's comment.

[Figure]

[Figure]

**Figure 7: Time series of the frequency of occurrence, expressed in terms of number of days per winter season, of the six synoptic types (ST) emerged from the cluster analysis, i.e. ST1 (a), ST2 (b), ST3 (c), ST4 (d), ST5 (e) and ST6 (f). In all panels, the missing data are highlighted as yellow bars, whereas the red line shows the 20-year moving average smooth. The period from 1884 to 2015 has been considered. The red line shows the 20-years moving average smoothing. On all panels, the vertical black lines define the limits of the 20-years sub-periods introduced in Section 3.1, except for the last period, which reduces to 2004/05 to 2014/15.**

**Technical corrections**

**RC (1):** Line 39 – the mid-1970s.

**AR (1):** Ok, we have corrected, thank you.

**RC (2):** Line 43 – the Castilla y Leon region

**AR (2):** Ok, we have corrected, thank you.

**RC (3):** Line 54 – few studies extended their analyses further back (?)

**AR (3):** Ok, we have corrected, thank you.

**RC (4):** Line 61 – to provide

**AR (4):** Ok, we have corrected, thank you.

**RC (5):** Line 89 – emphasized

**AR (5):** Ok, we have corrected, sorry for the mistake.

**RC (6):** Line 93 – to point

**AR (6):** Ok, we have corrected, thank you.

**RC (7)**: Line 155 – ascending order

**AR (7):** Ok, we have corrected, thank you.

**RC (8):** Line 280 – where the air pressure is lower than.

**AR (8):** Ok, we have corrected, thank you.

**RC (9):** Line 428 – I would write XX as 20th here.

**AR (9):** Ok, we have corrected, thank you.

**RC (10):** Line 479 – to inspect

**AR (10):** Ok, we have corrected, thank you.

**RC (11):** Line 557 – left for future studies

**AR (11):** Ok, we have corrected, thank you.

**RC (12):** Figure 2 – is there any way to increase the resolution of the photo? I can't really read it even zooming in and would really like to see what the records look like!

**AR (12):** Ok, we tried to increase the definition of this figure. Note that, according to the suggestions of the referee #1, we have moved this figure into the Appendix A. Therefore, in the new manuscript version this figure will be labelled as Figure A1.

**Figure A1.** This picture presents an example of the original data source (related to the second 10-day period (i.e. from day 11 to 20) of January 1946). Each row accounts for the observations of a specific day, and each column is devoted to the records of a determined parameter at a specific hour of the day. The columns including snowfall measurements are bordered with red lines.

**RC (13):** Figure 3 – it's pretty hard to find the panel labels in this figure.

**AR (13):** Ok, we have corrected, thank you. Note that this figure is now labelled as Figure 2.

[Figure]

**Figure 2: Histograms of daily height of snow (HNS$_d$) observed in MVOBS in December (a), January (b) and February (c). The y-axis is the absolute frequency (expressed as number of days), whereas the x-axis is the HNS$_d$ amount at the bin centre (in cm). Data collected between 1884 and 2020 have been taken into account.**

**RC (14):** Figures 6 and 7 – Beautiful. Is it possible to project these data so the higher latitude portions take up less of the map? I understand this can be a huge headache, however.

**AR (14):** The maps sketched in Figures 6 and 7 (numbered as Figure 5 and 6 in the revised manuscript) have been elaborating in MATLAB using the Miller Cylindrical projection. We have tried all other projection options offered by the MATLAB toolbox m_map (https://www.eoas.ubc.ca/~rich/mapug.html#p29): the only good alternative is the Equidistant cylindrical projection, which consists of equally-spaced latitude and longitude lines. We hope that this solution met the referee's requirement.

[revised manuscript text omitted]

---

## Referee Report (RR1)

The authors are to be commended for the effort that they have put into revising the manuscript. They addressed all comments and included all my suggestions. Thank you. The manuscript has considerably improved.

One thing I still do not agree with is the Hovmöller-type diagram. Please correct me if I'm wrong, but I think you have annual values for all five teleconnection indices? So what is shown between, e.g., AO and EAWR for one year? For me it looks like a gradient between the two winter values of AO and EAWR, but this does not make sense. I agree with the authors, that the plot could serve as a nice overview between all teleconnection indices and HNS, but I think it's technically wrong and thus potentially misleading. You can still keep the type of plot, but need to separate the x-axis into a discrete one. See also the figure below (with some random values):

[Figure]

Otherwise, there are only a few minor technical things left:
- Regarding the monthly sums: Did you calculate them only when all (28-31) daily observations were available? Did you not have any gaps of few days in between? Usually, one calculates monthly means, if 90% or 95% of daily observations are available.
- New Figure 4: the top right panel should be "moving average" without "standard deviation", no?
- Table 3: Please add one column with the correlations using all years.
- All correlation tables: The choice of using a soft grey colour for denoting $0.05 < p < 0.1$ is unlucky, since the natural visibility order would be **bold**, normal, grey. I think one p-value threshold is enough. For correlations 0.1 (90%) is well accepted, especially since you have only 20 observations (time periods) in some cases. Then you can use bold for significant and normal for not. You can use 0.1 (90%) consistently for all correlation tables (now you have sometimes only 0.05, and sometimes both 0.05 and 0.1).
- L395ff: Again, t-test is standard, so no need to specify hypotheses. But the t-test is a pair comparison. From the response to the comment to referee 2, I understand what you did, but not from what you wrote in the manuscript. Maybe you can clarify into something like "HNS was significantly different between the high (STx, STy, Stz) and low group (ST...) ($p<0.05$, t-test)."

---

## Author Response (AR2)

Revision of

**"Synoptic control over winter snowfall variability observed in a remote site of Apennine Mountains (Italy), 1884–2015"**
*V. Capozzi, C. De Vivo, G. Budillon*

**RC** = Referee comment
**AR** = Authors' reply
Note that in the new manuscript version, all changes are marked in yellow.

**REVIEWER #1**

**RC (1)**: The authors are to be commended for the effort that they have put into revising the manuscript. They addressed all comments and included all my suggestions. Thank you. The manuscript has considerably improved. One thing I still do not agree with is the Hovmöller-type diagram. Please correct me if I'm wrong, but I think you have annual values for all five teleconnection indices? So what is shown between, e.g., AO and EAWR for one year? For me it looks like a gradient between the two winter values of AO and EAWR, but this does not make sense. I agree with the authors, that the plot could serve as a nice overview between all teleconnection indices and HNS, but I think it's technically wrong and thus potentially misleading. You can still keep the type of plot, but need to separate the x-axis into a discrete one. See also the figure below (with some random values):

[Figure]

**AR (1):** Dear Reviewer, we are very grateful for his/her positive evaluation about our revision work. We are also grateful for your further comments, which help us to improve our paper.
We agree with your remarks about the Fig. 10 and we have modified it following your precious suggestions (see below). Note that we have applied a light smoothing in the y direction.

[Figure]

**Figure 10**: The left panel presents the behaviour in time of the teleconnection indices adopted in this study. More specifically, in the x-axis the winter value of Arctic Oscillation (AO), Eastern Atlantic Western Russia (EAWR), Eastern Mediterranean Pattern (EMP), North Atlantic Oscillation (NAO) and Scandinavia pattern (SCAND) is show from left to right. The indices values are color-coded according to the horizontal bar and are expressed as standardized units. The right panel shows the winter (December to February) time series of total height of new snow, expressed as anomalies (in cm) with respect to 1981-2010 period. Red bars highlight winters with positive anomaly (in cm), blue bars winters with negative anomaly. The missing data are marked as yellow bars. For both panels, the period 1950/51-2014/15 is considered.

**RC (2):** Regarding the monthly sums: Did you calculate them only when all (28-31) daily observations were available? Did you not have any gaps of few days in between? Usually, one calculates monthly means, if 90% or 95% of daily observations are available.

**AR (2):** We have calculated the monthly values if the 95% of daily observations are available. We have clarified this point in the revised manuscript (See Line 140).

**RC (3):** New Figure 4: the top right panel should be "moving average" without "standard deviation", no?

**AR (3):** Yes, we apologize for the typo. We have corrected the legend of this figure.

**RC (4)**: Table 3: Please add one column with the correlations using all years.

**AR (4):** Ok, we added a column in Table 3 with the correlation values computed over all considered time interval.

**RC (5)**: All correlation tables: The choice of using a soft grey colour for denoting $0.05 < p < 0.1$ is unlucky, since the natural visibility order would be bold, normal, grey. I think one p-value threshold is enough. For correlations 0.1 (90%) is well accepted, especially since you have only 20 observations (time

periods) in some cases. Then you can use bold for significant and normal for not. You can use 0.1 (90%) consistently for all correlation tables (now you have sometimes only 0.05, and sometimes both 0.05 and 0.1).

**AR (5):** According to the referee's suggestion, for all correlation tables (i.e. Table 3, Table 4 and Table 5), we have considered 0.1 (90%) as reference p-value for significance test and we used bold to indicate the significant values and normal for not significant ones.

**RC (6):** L395ff: Again, t-test is standard, so no need to specify hypotheses. But the t-test is a pair comparison. From the response to the comment to referee 2, I understand what you did, but not from what you wrote in the manuscript. Maybe you can clarify into something like "HNS was significantly different between the high (STx, STy, Stz) and low group (ST...) (p<0.05, t-test)."

**AR (6):** In the revised manuscript version, we have better clarified this point as follows (see Lines 396-398 in the main text): **"**It should be pointed out that the average HNSd found for ST1, ST2 and ST4 significantly differ from the HNSd values found for ST3, ST5 and ST6 (i.e. according to the t-test, p<0.05 for all ST pairs, for example ST1 vs. ST5 and ST2 vs. ST6)."

**REVIEWER #2**

**RC:** The revised manuscript addresses the primary concerns of both reviewers. I particularly appreciate the authors' inclusion of a more explicit discussion section and manner in which they have handled the suggestions to revise their sub-period selection (Sec. 3.1). Additionally, the delineation of the sub-periods in the figures really helped me with the interpretation of the authors' results. Personally, I really appreciated the higher resolution photo presented in Figure A1 -- what a cool snow data source rich in human history!

The revised manuscript more adequately contextualizes the results in a broader cryosphere perspective, while also employing more robust methods. I believe a broad audience of TC readers will enjoy and take interest in this work.

I would suggest some technical corrections to the language in the manuscript prior to publication.

**AR:** Dear Dr. Hancock, we are very grateful for your positive comments about our revision work. We thank you again for all suggestions and remarks and for the time you spend for the revision of our work. Thank you in advance for the technical language corrections.

Best regards.